# Restoration of retinal regenerative potential of Müller glia by disrupting intercellular Prox1 transfer

Eun Jung Lee [1,2,3,9], Museong Kim [1,2,9], Sooyeon Park[1,2,3], Ji Hyeon Shim[3], Hyun-Ju Cho[4,5], Jung Ah Park[3], Kihyun Park[1], Dongeun Lee [1,2], Jeong Hwan Kim[6], Haeun Jeong [1,2], Fumio Matsuzaki [7,8], Seon-Young Kim [6], Jaehoon Kim[1], Hanseul Yang [1,2], Jeong-Soo Lee [4,5] & Jin Woo Kim [1,2,3] ✉

Individuals with retinal degenerative diseases struggle to restore vision due to the inability to regenerate retinal cells. Unlike cold-blooded vertebrates, mammals lack Müller glia (MG)-mediated retinal regeneration, indicating the limited regenerative capacity of mammalian MG. Here, we identify prospero-related homeobox 1 (Prox1) as a key factor restricting this process. Prox1 accumulates in MG of degenerating human and mouse retinas but not in regenerating zebrafish. In mice, Prox1 in MG originates from neighboring retinal neurons via intercellular transfer. Blocking this transfer enables MG reprogramming into retinal progenitor cells in injured mouse retinas. Moreover, adeno-associated viral delivery of an anti-Prox1 antibody, which sequesters extracellular Prox1, promotes retinal neuron regeneration and delays vision loss in a retinitis pigmentosa model. These findings establish Prox1 as a barrier to MG-mediated regeneration and highlight anti-Prox1 therapy as a promising strategy for restoring retinal regeneration in mammals.

Cells that constitute animal tissues are produced during development and subsequently degenerate over an organism's lifetime. In many animal tissues that contain stem cells, such as skin, liver, and intestines, degenerated cells are replenished by new cells, allowing these tissues to maintain their functions despite continuous cell loss[1]. However, neurons in most tissues within the mammalian central nervous system (CNS) lack the capacity for regeneration[2], whereas CNS tissues in cold-blooded vertebrates, such as fish and amphibians, retain their regenerative potential[3,4]. Previous studies have suggested that this discrepancy arises from the absence of promoting factors and/or the presence of suppressive factors that regulate a series of regenerative events, including the reprogramming, proliferation, and neuronal

differentiation of neural stem/progenitor cells (NSCs/NPCs). However, the identities of these regulatory factors remain largely unknown.

The retina is a valuable model for studying neural tissue development and regeneration mechanisms. Retinal regeneration primarily depends on Müller glia (MG), which can be reprogrammed into neural progenitor cells (MG-derived retinal progenitor cells, or MGPCs) that re-enter the cell cycle to regenerate retinal neurons[3–5]. MG are known to have proliferative potential in adult fish retinas and are thought to be a source of retinal neurons, giving rise after injury-induced degeneration[6,7]. MGPCs re-enter the cell cycle to expand themselves and differentiate into various retinal cell types, including rod photoreceptors (rPRs) and retinal ganglion cells (RGCs), which are affected

[1]Department of Biological Sciences, Korea Advanced Institute of Science and Technology (KAIST), Daejeon, South Korea. [2]KAIST Stem Cell Research Center, Korea Advanced Institute of Science and Technology (KAIST), Daejeon, South Korea. [3]Celliaz Ltd., Daejeon, South Korea. [4]Microbiome Convergence Research Center, Korea Research Institute of Bioscience and Biotechnology, Daejeon, South Korea. [5]KRIBB School, University of Science and Technology, Daejeon, South Korea. [6]Korea Bioinformation Center, Korea Research Institute of Bioscience and Biotechnology, Daejeon, South Korea. [7]Laboratory for Cell Asymmetry, RIKEN Centre for Biosystems Dynamics Research, Kobe, Hyogo, Japan. [8]Present address: Department of Aging Science and Medicine, Graduate School of Medicine, Kyoto University, Kyoto, Japan. [9]These authors contributed equally: Eun Jung Lee, Museong Kim. ✉e-mail: jinwookim@kaist.ac.kr

in retinal diseases such as retinitis pigmentosa (RP) and glaucoma, respectively. This differentiation is tightly regulated by transcription factors and extracellular signals, which help that regenerated neurons integrate into the existing retinal circuitry.

The regeneration process in fish and amphibians is initiated by signals from the damaged retina, where multiple pathways regulate MG reprogramming and proliferation. For example, Wnt signaling is activated in response to retinal damage and its inhibition prevents regeneration, although it alone is insufficient to regenerate retinal neurons but can reprogram MG to RPC[8]. Notch signaling is also activated during zebrafish retina regeneration[9,10], however it should be suppressed for MGPC proliferation and their subsequent neuronal differentiation[11,12]. In contrast, sonic hedgehog (Shh) signaling, which shares hairy and enhancer of split-1 (Hes1) with Notch signaling for a downstream effector in RPCs[13], could promote MGPC proliferation and guide the differentiation of MGPCs into retinal neurons[14]. Other mitogens, such as fibroblast growth factor (Fgf) and heparin-binding epidermal growth factor (Hbegf), are also upregulated following retinal damage and promote MGPC proliferation and neural differentiation in both fish and amphibians[15]. In addition to these external cues, changes in intracellular pathways that suppress yes-associated protein (Yap) and WW domain-containing transcription regulator 1 (TAZ) can also support this process[16,17].

However, unlike in cold-blooded vertebrates, mammalian MG cells do not naturally reprogram into RPCs capable of regenerating retinal neurons after injury. Instead, they partially dedifferentiate before returning to a quiescent state[18,19]. This limited regenerative potential has been attributed to the lack of activating factors and the presence of inhibitory factors. For example, the regenerative incompetence of mammalian MG involves the defective expression of key neurogenic factors, such as *achaete-scute family bHLH transcription factor 1* (*Ascl1*) and *lin-28*[18,20,21], as evidenced by the ability of ectopically expressed Ascl1 to trigger the reprogramming of MG into RPCs in the injured mouse retina[22]. However, Ascl1 alone is not sufficient; epigenetic changes, including histone acetylation and DNA demethylation, are also necessary for reprogramming MG into RPCs[23–25].

The Prospero-related homeobox 1 (Prox1), an evolutionarily conserved transcription factor, has been demonstrated to block the proliferation of NSCs/NPCs but induce the differentiation of these cells to the neurons[26,27]. In *Drosophila*, the Prox1 homolog *prospero* (*pros*) suppresses the expression of *achaete-scute complex* (*asc*) genes—homologous to mammalian Ascl1—and reduces *cyclin E* (*CycE*) levels while inducing *dacapo* (*DAP*), a cyclin-dependent kinase inhibitor, to prevent post-mitotic neuroblasts from re-entering the cell cycle[28]. These anti-NPC functions are also conserved in mammalian Prox1[29–31], where it plays a critical role in neurogenesis by facilitating cell cycle exit of NPCs in developing neural tissues[29–31].

In the developing mouse retina, Prox1 stimulates the differentiation of RPCs into horizontal cells (HCs)[31]. In the adult mouse retina, Prox1 is broadly expressed in the interneurons, including the HCs, bipolar cells (BCs), and A2 amacrine cell (AC) subset[31,32]. In this study, we found that, in addition to its expression in these retinal neurons, Prox1 is also present in the MG of the injured mouse retina but not in that of zebrafish. Notably, we discovered that Prox1 is not endogenously expressed in mouse MG; rather, it is acquired from retinal neurons through intercellular protein transfer—a known feature of homeodomain proteins (HPs)[33]. Accordingly, we could induce the reprogramming of MG to RPCs in the injured mouse retina by disrupting the transfer of Prox1 from donor retinal neurons to MG. Conversely, we suppressed MGPC proliferation in zebrafish by injecting recombinant Prox1 proteins intraocularly. These results suggest that exogenous Prox1 is a key suppressor of MG reprogramming; thus, its elimination should render mammalian MG competent for injury-induced retinal regeneration.

## Results
### Retinal injury-induced accumulation of Prox1 protein in mouse MG

In addition to its stable and robust expression in the mouse retinal neurons, Prox1 could be detectable in MG at relatively lower levels than those in the retinal neurons (Fig. 1a)[34]. Interestingly, Prox1 immunostaining signals in MG were significantly elevated following photoreceptor (PR) degeneration induced by N-methyl-N-nitrosourea (MNU)[35,36], whereas Prox1 intensities in other retinal subsets remained unchanged (Fig. 1a, c, d; Supplementary Fig. 1d). Similar findings were also observed in mouse retinas subjected to intensive light exposure

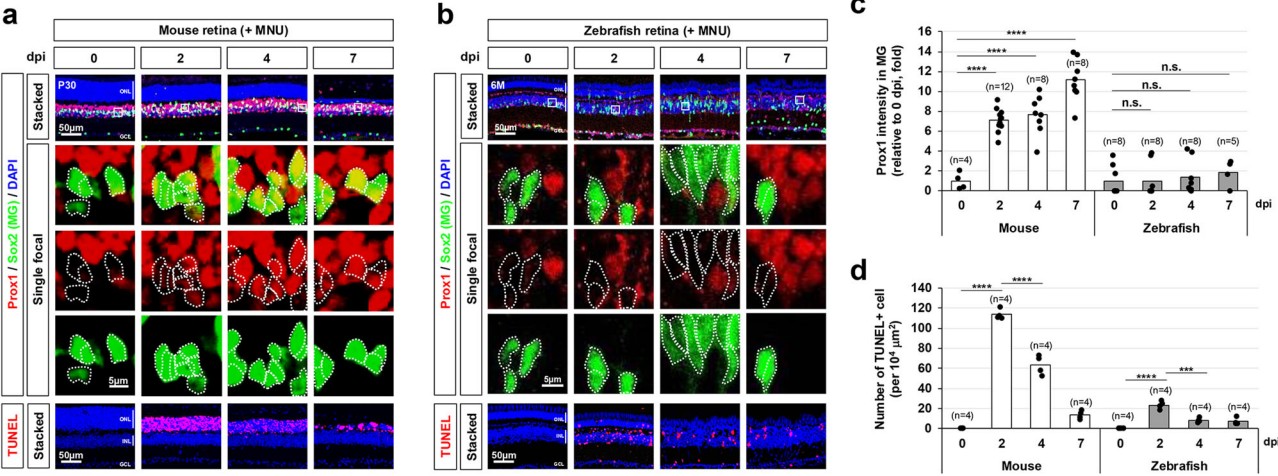

**Fig. 1 | Injury-induced accumulation of Prox1 in mouse MG. a, b** Retinal injury was induced in P30 mice (**a**) and 6-month-old zebrafish (**b**) through intraperitoneal injection of MNU, followed by immunostaining to assess Prox1 distribution in the retinas of both species. The bottom row displays single focal images, magnified from the boxed areas in the stacked images shown in the top row. Sox2-positive MG nuclei are outlined by dotted lines. Nuclei of the retinal cells are visualized by DAPI staining. **c** Relative Prox1 immunofluorescent intensity in MG at the indicated days post-injury (dpi), normalized to Prox1 intensity in BCs within the same image, is shown in the graph. Each dot represents the median intensity collected from one retina. Numbers of samples analyzed are shown in the graph (data from 4–6 independent litters). **d** Quantification of TUNEL-positive apoptotic cells in the specified retinal area. Columns represent mean values. Number of samples analyzed is 4. Statistical significance (p-values) of the data in (**c**) and (**d**) was calculated using one-sided Student's t-test (*, $p < 0.05$; **, $p < 0.01$; ***, $p < 0.005$; ****, $p < 0.001$; not significant (n.s.), $p > 0.05$).

that damages PRs, and intravitreal injection of N-methyl-D-aspartate (NMDA) that degenerates BCs, ACs, and RGCs[37] (Supplementary Fig. 1a, c, d). The Prox1 upregulation was also noted in *Crx-CreERT2;R26^DTA/+* mouse retinas, where selective PR degeneration is induced by diphtheria toxin A (DTA) (Supplementary Fig. 1b, d). In this model, tamoxifen (Tam), an estrogen analog, activates CreERT2 to drive DTA expression from the *ROSA26* (*R26*) gene locus in the CreERT2-expressing PRs[38,39]. The accumulation of Prox1 in MG was more pronounced in MNU-treated and *Crx-CreERT2;R26^DTA/+* mouse retinas compared to light-injured retinas, where PR degeneration and MG delamination were less severe (Fig. 1a, d; Supplementary Fig. 1a, d). These findings suggest that increased Prox1 expression in MG is a common response to retinal injury in mouse models, independent of the type of retinal cell degeneration but associated with the extent of injury. In contrast to the mouse retina, MNU- and NMDA-induced injuries did not significantly increase Prox1 levels in zebrafish MG (Fig. 1b, c; Supplementary Fig. 1, f–h), which are known to be reprogrammed to MGPCs in response to the retinal injuries for subsequent proliferation and differentiation to retinal neurons[3–5].

## Exogenous origin of Prox1 protein in mouse MG

The injury-induced accumulation of Prox1 in MG was an unexpected finding, as the levels of *Prox1* mRNA in MG lineage cells remained consistently low without significant alterations following NMDA- or light-induced injuries to mouse and zebrafish retinas[18] (Supplementary Fig. 2). To further explore this phenomenon, we sought to examine changes in *Prox1* mRNA expression in mouse MG using RNAscope in situ RNA hybridization. We, however, did not observe an increase of *Prox1* mRNA within the MG following the MNU injury (Fig. 2a, b). This result was further validated through quantitative reverse transcription-polymerase chain reaction (RT-qPCR) analysis of *Prox1* mRNA in tdTomato (tdTom)-positive MG, which were isolated from *Glast-CreERT;R26^tdTom/+* mouse retinas by fluorescence-activated cell sorting (FACS) (Fig. 2c). In this mouse model, CreERT is selectively expressed in MG by the *glutamate aspartate transporter* (*Glast*) promoter[40] and subsequently activated by Tam treatment to express the tdTom Cre reporter, enabling FACS analysis.

Subsequently, we aimed to investigate transcription at the mouse *Prox1* gene locus in MG by utilizing *Prox1^fg* mice, which are capable of expressing *EGFP* complementarily at the *Prox1* gene locus after the Cre-mediated deletion of the floxed *Prox1* knock-in allele[30] (Fig. 2d). As expected, robust EGFP expression was observed in HCs, BCs, and AC subsets in *Prox1^fg;Dkk3-Cre* mice, where the *Prox1* knock-in allele was eliminated in embryonic RPCs and their descendant retinal cells[41] (Supplementary Fig. 3, a–c). However, the complementary EGFP expression was scarcely observed in MG of the *Prox1^fg;Dkk3-Cre* mouse retinas, whereas EGFP expression was found in their retinal neurons (Supplementary Fig. 3a, c).

In the Tam-treated *Prox1^fg/fg;Glast-CreERT;R26^+/tdTom* mouse retinas, the tdTom-positive, Cre-affected, *Prox1*-deleted MG did not exhibit EGFP signals (Fig. 2e, f). Furthermore, EGFP signals did not show an increase in tdTom-positive MG following MNU- or NMDA-induced retinal injuries (Fig. 2f; Supplementary Fig. 4a, b). However, Prox1 immunostaining signals were still elevated in these MG cells in the injured *Prox1^fg/fg;Glast-CreERT;R26^+/tdTom* mouse retinas (Fig. 2e, g; Supplementary Fig. 4a, c). Taken together, our data suggest that MG in injured mouse retinas accumulate Prox1 without increasing the expression of endogenous *Prox1* gene.

## Exogenous Prox1 suppresses injury-induced proliferation of MG

Previously, we reported that Prox1 protein may have the potential to transfer between cells, a characteristic shared by the majority of homeodomain proteins[33]. Therefore, it is plausible that mouse MG could uptake Prox1 secreted from neighboring retinal neurons with active *Prox1* gene expression (i.e., HCs, BCs, and ACs) (Fig. 3a). Our data

demonstrate that BCs represent the largest retinal cell population expressing the *Prox1* gene (Supplementary Fig. 3b). Thus, we investigated whether BCs could serve as a source of Prox1 for MG by genetically eliminating the *Prox1* gene and expressing *EGFP* complementarily in BCs of *Prox1^fg/fg;Chx10-CreERT2* mice, which express CreERT2 predominantly in the BC population in a Tam-dependent manner[30,42] (Fig. 3b). This manipulation led to a significant reduction in Prox1 intensity in EGFP-negative MG as well as EGFP-positive BCs in *Prox1^fg/fg;Chx10-CreERT2* mouse retinas compared to those in *Prox1^fg/fg* littermate mouse retinas, without changes in Prox1 intensity in HCs and ACs (Fig. 3c, d; Supplementary Fig. 5a, b). However, the numbers of BCs or other retinal cells in *Prox1^fg/fg;Chx10-CreERT2* mice relative to the *Prox1^fg/fg* littermates did not show significant differences (Supplementary Fig. 5a, c). The visual acuity of the *Prox1^fg/fg;Chx10-CreERT2* mice was not also significantly different from that of the *Prox1^fg/fg* littermates (Supplementary Fig. 5d).

Remarkably, the number of EdU-labeled MG significantly increased in the injured retinas of *Prox1^fg/fg;Chx10-CreERT2* mice (Fig. 3, e–g). These findings suggest that MG gain proliferative potential upon the decrease of exogenous Prox1 transferred from retinal neurons. To test this hypothesis, we supplied Prox1 to the retina by intravitreal injection of FLAG-tagged recombinant Prox1 protein to *Prox1^fg/fg;Chx10-CreERT2* mouse eyes (Fig. 3b). The FLAG-Prox1 failed to penetrate vehicle-treated uninjured mouse retinas, which maintained an intact inner retina-blood barrier (Supplementary Fig. 6). However, in MNU-injured retinas, FLAG-Prox1 proteins were detectable in MG, with their presence persisting for several days. In contrast, FLAG peptides did not become entrapped in the retina due to their lack of affinity for cell surfaces, a characteristic typical of HPs[33]. The delivery of FLAG-Prox1 resulted in a decrease in the number of EdU-labeled MG but an elevation in the number of EdU-labeled microglia in the injured retinas of *Prox1^fg/fg;Chx10-CreERT2* mice (Fig. 3, e–g).

To further investigate the inverse relationship between the exogenous Prox1 protein accumulation and MG cell proliferation, we also injected FLAG-Prox1 into zebrafish eyes (Supplementary Fig. 7a), which express *glial fibrillary acidic protein* (*gfap*)-*EGFP* transgene in their MG[43]. The FLAG-Prox1 was detectable in EGFP-positive MG of MNU-injured *gfap:EGFP* zebrafish retinas (Supplementary Fig. 7b, c). This manipulation led to a decrease in the number of Ascl1a-positive MGPCs in these fish retinas (Supplementary Fig. 7d, e). It also reduced the EdU-positivity of Sox2;EGFP-positive MG or MGPCs (Supplementary Fig. 7f, k, l). Consequently, the regeneration of retinal neurons, identified by their incorporation of EdU and loss of MG-specific gfap-EGFP expression but gain of cell type-specific marker expression, was diminished in the injured zebrafish retinas injected with FLAG-Prox1 (Supplementary Fig. 7, g–l).

## Recovery of MG proliferation in injured mouse retina by sequestering extracellular Prox1

Alternatively, we also aimed to disrupt Prox1 transfer to MG by sequestering it in the extracellular space using anti-Prox1 antibody (Fig. 4a). The rabbit polyclonal anti-Prox1 antibody that was injected into the intravitreal space of MNU-injured *Glast-CreERT;R26^tdTom/+* mouse eyes could reduce Prox1 level in MG, suggesting the blockage of intercellular transfer of Prox1 by the antibody (Supplementary Fig. 8). Further, for stable delivery of the antibody to the retina, we prepared a cDNA encoding a single-chain fragment (scFv) of chicken immunoglobulin with an affinity for Prox1 (Supplementary Fig. 9a). The anti-Prox1 scFv antibody (αProx1) effectively blocks the transfer of Prox1 between cultured cells when added to the growth medium (Supplementary Fig. 9b, c). The αProx1 was fused with the signal sequence of human interleukin-2 (IL-2) before introduction into the genome of adeno-associated virus (AAV), which is known for its effective gene delivery to the mouse retina[44]. The resulting AAV2/2 serotype viruses (AAV2-αProx1) were injected into the vitreal cavity of mouse eyes,

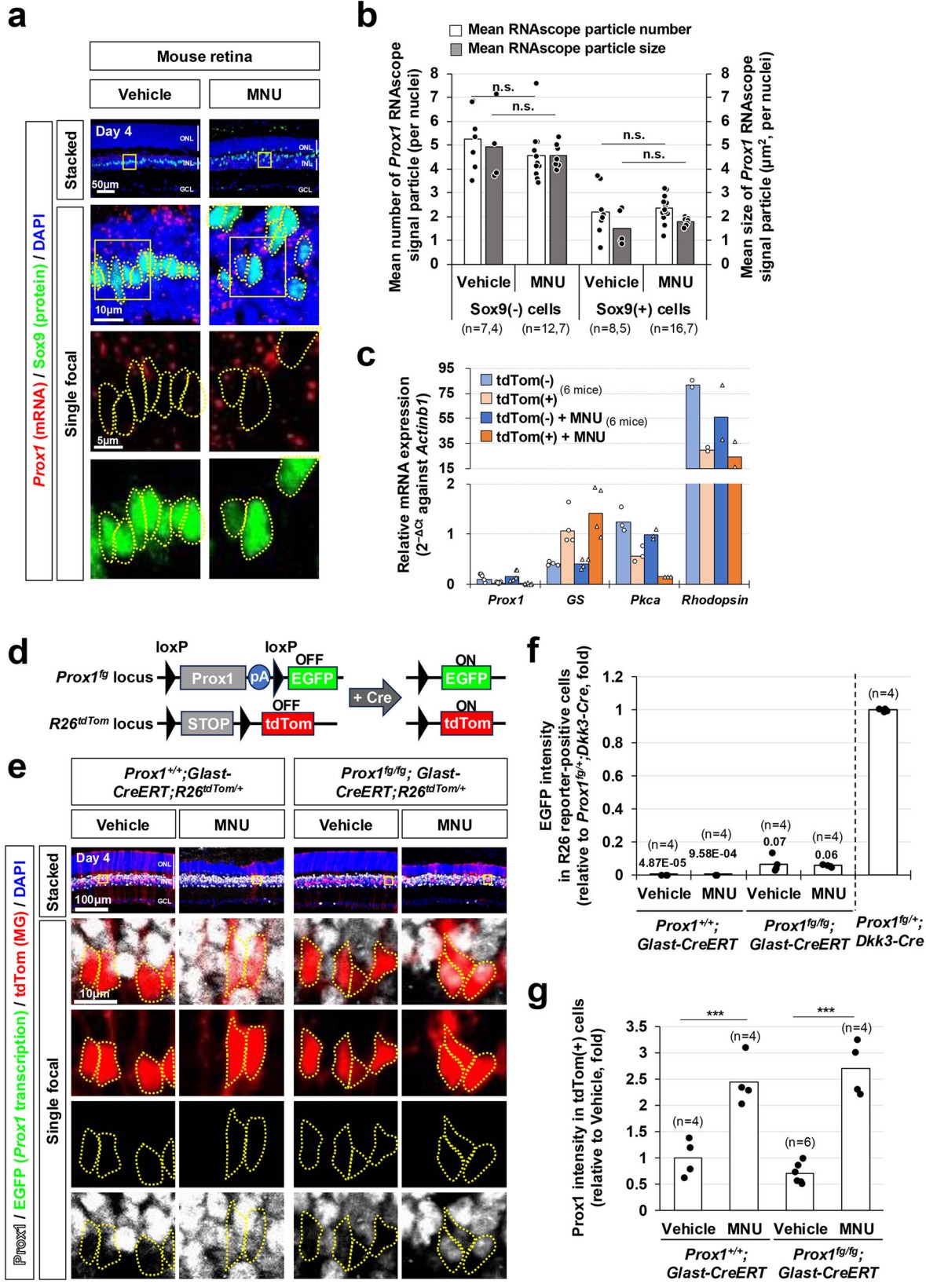

targeting retinal cells in the GCL and INL to secrete αProx1 into the extracellular space of the retina from the infected cells (Fig. 4b, c; Supplementary Fig. 10a, c). In the mouse retinas infected with AAV2-αProx1, the injury-induced accumulation of Prox1 in the MG was inhibited without altering Prox1 levels in retinal neurons (Fig. 4c, d; Supplementary Fig. 10b). In contrast, infection with AAV2-Control

antibody (AAV2-Ctrl Ab), expressing an scFv with no affinity to Prox1, had no such effect. Furthermore, we noted that the number of MG incorporating EdU for 4 days post-MNU injury was elevated in mouse retinas infected with AAV2-αProx1 compared to those infected with AAV2-Ctrl Ab (Fig. 4, e–g). Taken together, our findings suggest that Prox1 originated from retinal neurons, including BCs, exerts a

**Fig. 2 | Exogenous origin of Prox1 in mouse MG. a** *Prox1* mRNA in mouse retinal sections was visualized using RNAscope, combined with immunostaining for the MG marker Sox9. **b** Mean number and size of *Prox1* RNAscope signals in MG nuclei and the rest INL cell nuclei are presented in the graph. Numbers of samples analyzed are shown in the graph (data from 3 independent litters). **c** tdTom-positive and tdTom-negative retinal cells were isolated from *Glast-CreERT;R26+/tdTom* mouse retinas by FACS. Relative mRNA expression of indicated genes to *Actinb1* was determined by their cycle threshold (Ct) scores obtained using RT-qPCR (data from 4 independent batches). **d, e** Complementary EGFP expression from the mouse *Prox1* gene locus, following Cre-dependent excision of Prox1 cDNA, was assessed by co-immunostaining mouse retinal sections with chicken anti-GFP and rabbit anti-Prox1 antibodies. Cre activity in the same retinal areas was verified by expression of

R26^tdTom reporter. tdTom-positive cell bodies are outlined by dotted-lines. pA, polyA transcription terminator. **f** EGFP intensity in tdTom-positive cells was measured by confocal microscopy. The relative intensity, normalized to the EGFP intensity of cells in *Prox1fg/+;Dkk3-Cre* mouse retinas (data shown in Supplementary Fig. 3a), is plotted. Numbers of samples analyzed are shown in the graph (data collected from 3 independent litters). The mean values are indicated in the graph. **g** Relative Prox1 immunofluorescent intensity in MG in the corresponding retina, normalized to Prox1 intensity in BCs within the same image, is plotted. Each dot represents the median intensity collected from one retina. Numbers of samples analyzed are shown in the graph (data from 3 independent litters). *P*-values in the graphs were calculated using one-sided Student's t-test (n.s., *p* > 0.05; ***, *p* < 0.005).

suppressive role in injury-induced proliferation of MG and/or transition to RPCs through intercellular transfer.

## Transition of Prox1-depleted MG into RPCs in the injured mouse retina

Next, we investigated whether the identity transition of MG to RPCs really occurred in MNU-injured *Prox1fg/fg;Chx10-CreERT2* mouse retinas through single-cell RNA sequencing (scRNA-seq) analysis (Supplementary Fig. 11–13; see details in Methods). MG populations in MNU-injured mouse retinas could be categorized into three clusters (Fig. 5a, b). Cluster #3 appeared in both healthy and injured mouse retinas, suggesting it represents resting MG population (Supplementary Fig. 12a, b). In contrast, clusters #1 and #7, representing activated MG populations, were selectively enriched in the injured mouse retinas (Fig. 5a, b; Supplementary Fig. 12a, b). This was supported by the cell lineage flow from cluster #3 to cluster #7 in the velocity plot of scRNA-seq data (Fig. 5c).

We also observed a distinctive lineage to the BC (#6) and AC (#14) clusters from cluster #20 (Fig. 5c), which was detected at significant cell numbers only in the injured *Prox1fg/fg;Chx10-CreERT2* mouse retinas (Fig. 5d; Supplementary Fig. 12d, e). This suggests a potential progenitor cell identity for this cluster. Supporting this, cluster #20 expressed RPC markers, *growth arrest and DNA damage alpha and gamma (Gadd45a and Gadd45g)*[45] (Fig. 5e; Supplementary Fig. 13a). Additionally, this cluster expressed *heparin-binding epidermal growth factor (Hbegf)* and its downstream *early growth response protein 1 (Egr1)*, which are involved in MG-to-RPC reprogramming[15,46], although it did not contain *Ascl1*, a key regulator of MG reprogramming[20,22] (Fig. 5e; Supplementary Fig. 13a, b).

Notably, cluster #20 contained cells expressing *hairy and enhancer of split-1 (Hes1)* and its upstream *Notch1*, both activated during MG-to-RPC reprogramming[15,46,47] (Fig. 5e). Hes1 expression, which is typically low in resting MG, peaked at 2 days post-MNU injury in Sox2-positive MG or MGPCs in *Prox1fg/fg;Chx10-CreERT2* mouse retinas, while it declined in *Prox1fg/fg* mouse retinas (Fig. 5f, g; Supplementary Fig. 14a). These results suggest that elevated Notch1 in MG of MNU-injured *Prox1fg/fg;Chx10-CreERT2* mouse retinas may suppress *Ascl1* expression via Hes1 while other RPC markers are expressed in these cells.

Cells in cluster #20 also expressed *E2F transcription factor 5 (E2f5)*, *Myc*, and *proliferating cell nuclear antigen (Pcna)* (Fig. 5h; Supplementary Fig. 13c), markers enriched in proliferative RPCs[18]. Additionally, *Cyclin d1 (Ccnd1)* and *e1 (Ccne1)*, negative transcription targets of Prox1 in NSCs[28,48,49], along with *Cdk4*, which forms a complex with Ccnd1 to trigger G1-to-S cell cycle progression[50,51], were detected in cluster #20 cells (Fig. 5h; Supplementary Fig. 13d). Ccnd1-expressing MG or RPCs were elevated and sustained in the MNU-injured retinas of *Prox1fg/fg;Chx10-CreERT2* mice, whereas they appeared only transiently in *Prox1fg/fg* mouse retinas (Fig. 5i,j; Supplementary Fig. 14b). Together, these results suggest that cluster #20 represents proliferative RPCs.

We further analyzed gene expression changes selectively in MG cell lineages. tdTom-labeled MG and their descendant cells were

isolated by FACS from MNU-injured or uninjured *Glast-CreERT;R26tdTom/+* mouse retinas, which were infected with either AAV2-Ctrl Ab or AAV2-αProx1, and mRNA expression in individual cells was assessed using SMART (Switching Mechanism At the 5' end of RNA Template) sequencing (Fig. 6a; see details in Methods). We could identify a lineage flow from resting MG to RPC-like cells via activated MG, along with pseudotime progression (Fig. 6b, c). Proliferating cell markers *E2f5*, *Pcna*, *Cdk4*, and *Ccnd1*, as well as MGPC markers, including *Gadd45a*, *Hbegf*, *Notch1*, and *Hes1*, were enriched in late pseudotime cells, particularly in MNU-injured retinas infected with AAV2-αProx1 (Fig. 6d, e). Consistently, the tdTom-positive MG lineage cells co-expressing Hes1 and Ccnd1 were increased at significant levels in AAV2-αProx1-infected mouse retinas upon MNU injury (Fig. 6d–h).

Collectively, our findings suggest that exogenous Prox1 may inhibit the conversion of MG into MGPCs, potentially by suppressing the expression of its target genes, such as *Notch1* and *Ccnd1*[28,29]. This effect may also occur indirectly via the regulation of MG reprogramming factors, including *Hbegf*, which has been shown to induce the expression of *Notch1* and *Ccnd1*[15]. Therefore, inhibiting Prox1 transfer to MG appears necessary for their reprogramming into MGPCs, thereby restoring regenerative potential in the injured retina (Fig. 6i).

## Delaying vision loss in early-onset retinitis pigmentosa mouse models by blocking Prox1 transfer

Despite the development of various therapeutic interventions for retinal degenerative diseases, the majority of patients cannot recover their vision as their lost retinal cells are not regenerated. Therefore, the blocking PROX1 transfer could be a potent therapeutic strategy for retinal degenerative diseases, if exogenous PROX1 also plays the suppressive roles in human retinal regeneration. Indeed, PROX1 was accumulated in SOX2-positive MG of 79 years-old retinitis pigmentosa (RP) patient retina but not in the MG of 83 years-old healthy donor retina (Fig. 7a–c). The elevation of Prox1 was also observed in the MG of *Pde6brd10/rd10* mice (Fig. 7d, e), which carry retinal dystrophy-10 (rd10) mutation in the phosphodiesterase 6b gene (*Pde6b*) and lose their vision due to the degeneration of rod photoreceptors (rPRs), followed by the loss of cone photoreceptors (cPRs)[52,53].

Thus, we assessed the therapeutical potential of αProx1 in retinal degenerative diseases by infecting *Pde6brd10/rd10* mouse retina with AAV2-αProx1. We observed a significant decrease in Prox1 intensity in MG of *Pde6brd10/rd10* mouse retinas infected with AAV2-αProx1 compared to that in MG of AAV2-Ctrl Ab-infected *Pde6brd10/rd10* mouse retinas, whereas Prox1 intensity in BCs was unchanged (Supplementary Fig. 15a–c). Notably, the number of rPRs in the outer nuclear layer (ONL) in AAV2-αProx1-infected *Pde6brd10/rd10* mouse retinas was significantly increased in comparison to those in uninfected or AAV2-Ctrl Ab-infected *Pde6brd10/rd10* mouse retinas (Supplementary Fig. 15a, d). However, the numbers of TUNEL-positive apoptotic cells in AAV2-αProx1-infected *Pde6brd10/rd10* mouse retinas were not significantly different from those in the others (Supplementary Fig. 15a,e), indicating that the increase of the rPRs was not resulted from the decrease of cell death.

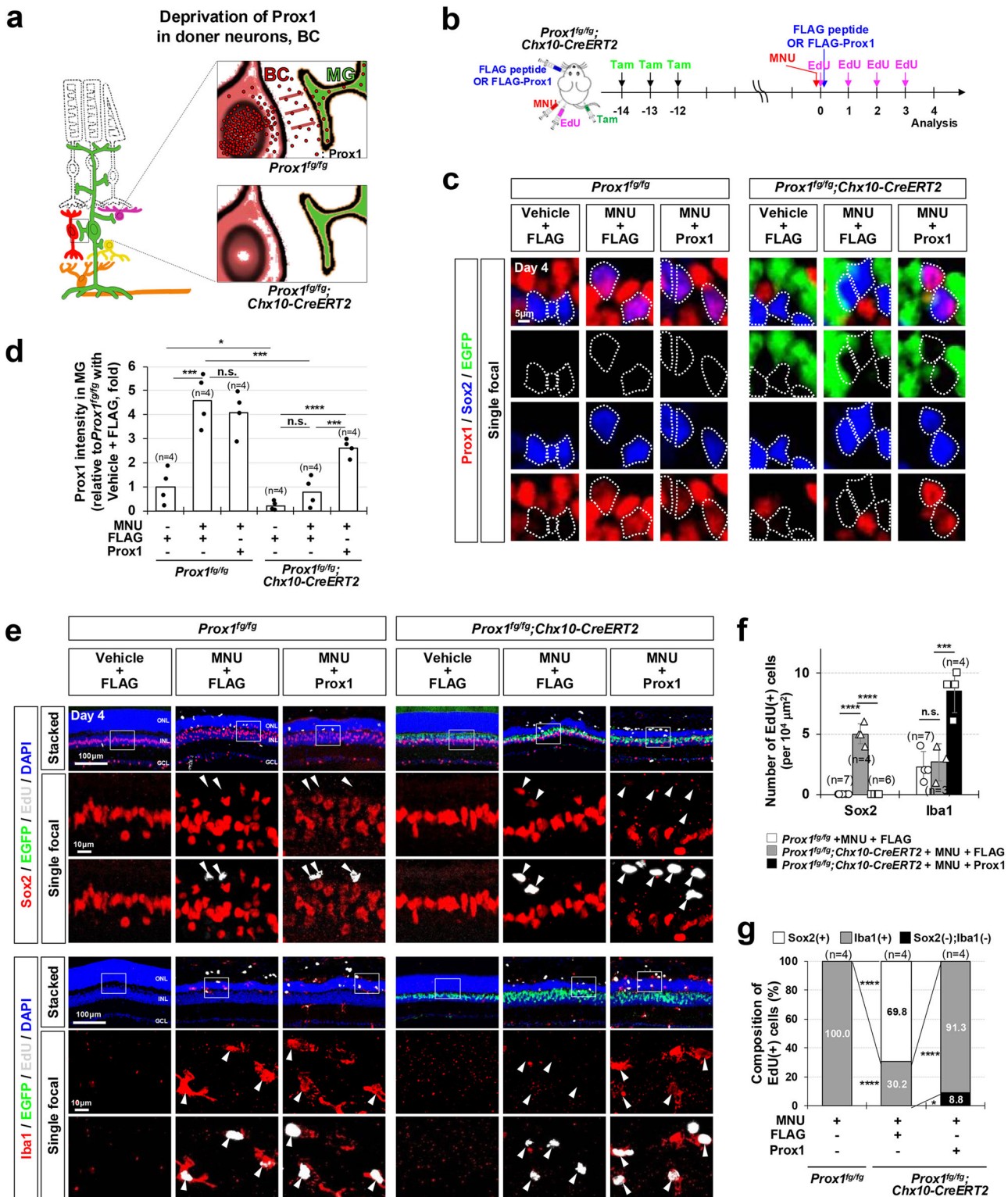

**Fig. 3 | *Prox1* gene deletion in donor retinal neurons restores MG proliferative potential. a** Schematic representation of Prox1 depletion in MG achieved through *Prox1* gene deletion in BC. (**b**) *Prox1* was selectively deleted and EGFP was complementarily expressed in BCs of *Prox1^{fg/fg};Chx10-CreERT2* mouse retinas by repeated Tamoxifen (Tam) injections. Following this, mice were injected with MNU to induce PR degeneration and EdU to label proliferating cells. As indicated, FLAG-Prox1 recombinant protein (250 fmol) or FLAG peptides were injected intravitreally. **c** Distribution of Prox1, Sox2, and EGFP in the retinas of *Prox1^{fg/fg}* and *Prox1^{fg/fg};Chx10-CreERT2* littermates was assessed by immunostaining. Sox2-positive MG nuclei are outlined by dotted-lines. **d** Relative Prox1 immunofluorescent intensity in MG in the corresponding retina, normalized to Prox1 intensity in BCs within the

same image, is shown. Each dot represents the median intensity collected from one retina. Number of samples analyzed is 4. **e** MG and microglial identities of EdU-labeled newborn cells in mouse retinas were determined by co-staining Sox2 and Iba1. The boxed areas in the top row are enlarged in the following two rows. Arrowheads point to EdU-labeled cell nuclei. **f** Quantification of EdU-labeled MG and microglia in the retinas is shown in the graph. **g** Composition of EdU-labeled cells in the mouse retinas is displayed in the graph. Numbers of samples analyzed are shown in the graph (data from 4 independent litters). Error bars denote SEM. *P*-values were calculated using one-sided Student's t-test (***, *p* < 0.005; ****, *p* < 0.001; n.s., > 0.05).

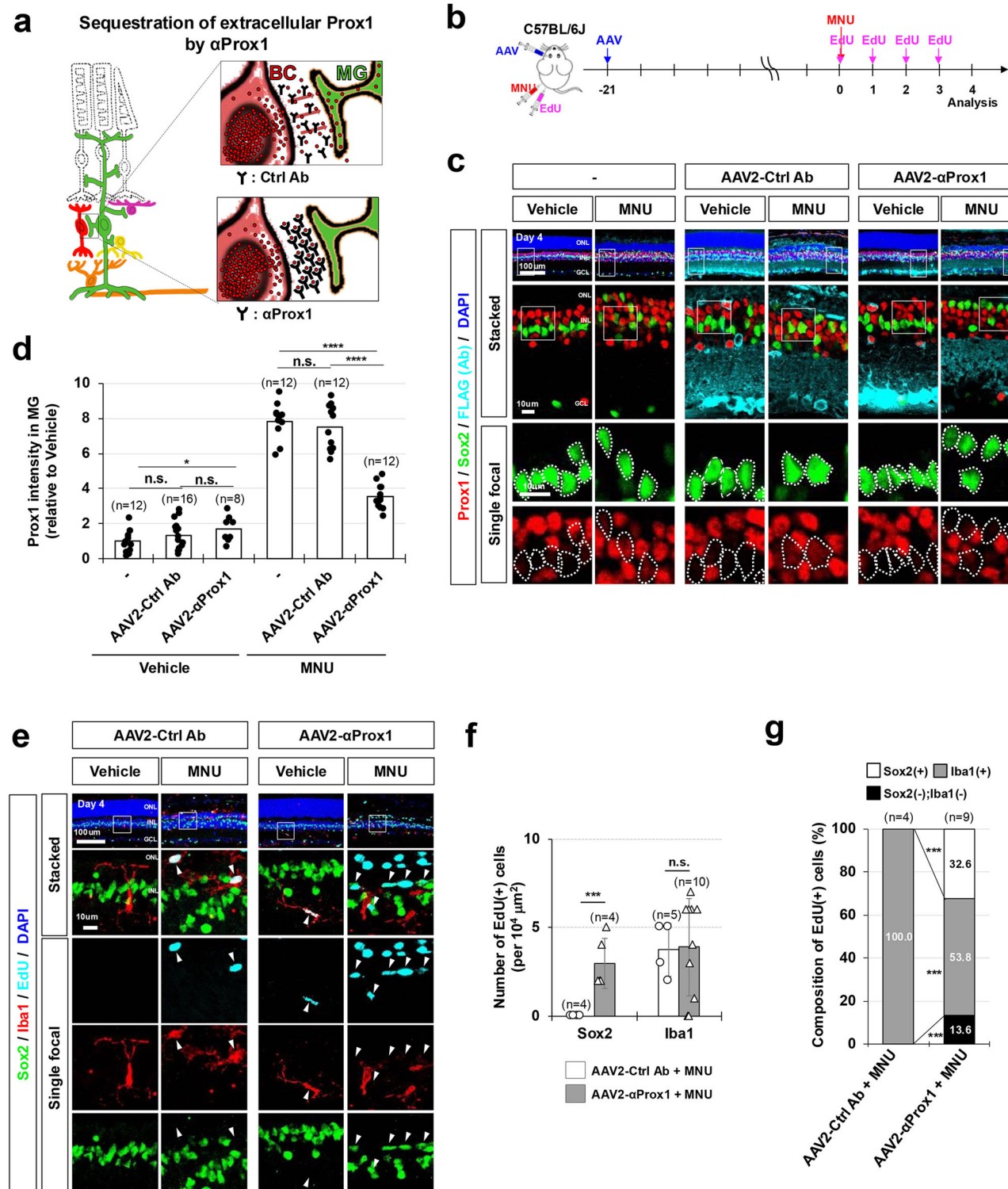

Thus, we investigated the increase of the ONL thickness was related with the regeneration of rPRs by identifying EdU-labeled newborn cells in the mouse retinas (Fig. 7f). EdU-labeled newborn Rhodopsin-positive rPRs were observed in the ONL of AAV2-αProx1-infected *Pde6b*^*rd10/rd10*^*;Glast-CreERT;R26t*^*dTom/+*^ mouse retinas, whereas no such cells were observed in AAV2-Ctrl Ab-infected mice (Fig. 7g, h). EdU incorporation was also observed in Sox2-positive MG or MGPCs and Pkcα-positive rod bipolar cells (rBCs) in those retinas, whereas the majority of EdU-labeled cells were microglia in AAV2-Ctrl Ab-infected samples (Fig. 7h; Supplementary Fig. 15f).

Furthermore, many of those EdU-labeled rPRs and rBCs in AAV2-αProx1-infected *Pde6b*^*rd10/rd10*^*;Glast-CreERT;R26*^*tdTom/+*^ mouse retinas expressed tdTom MG lineage cell marker (Fig. 7g, h; Supplementary Fig. 15f). Notably, due to the incomplete penetrance of Glast-CreERT activity in MG (Supplementary Fig. 15f), the tdTom-negative, EdU-labeled retinal neurons observed in these retinas could also have been derived from MG. Collectively, these findings suggest MG in AAV2-αProx1-infected *Pde6b*^*rd10/rd10*^ mouse retinas not only expanded themselves but also further differentiated into the retinal neurons.

**Fig. 4 | Sequestration of extracellular Prox1 by anti-Prox1 antibody restores MG proliferative potential. a** Schematic representation of Prox1 depletion in MG through sequestration of extracellular Prox1 protein by αProx1. **b** C57BL/6 J mice received intravitreal injections of AAV2-Ctrl Ab or AAV2-αProx1 at a concentration of $5 \times 10^9$ genome copies/eye. Mice were subsequently injected with MNU to induce retinal injury and EdU to label newly generated cells following MNU injury. **c** Retinas of the infected mice were subjected to immunostaining using mouse anti-FLAG antibody to detect FLAG-tagged proteins. Additionally, the distribution of Prox1 in the retinas was assessed by co-immunostaining with rabbit anti-Prox1 antibody. The MG identity of Prox1-positive cells was also determined through co-immunostaining with goat anti-Sox2 antibody. Enlarged views of boxed areas from the top row are presented in the second row and further magnified in the bottom rows, with Sox2-positive MG nuclei outlined by dotted lines. **d** Relative Prox1

immunofluorescent intensity in MG in the corresponding retina, normalized to Prox1 intensity in BCs within the same image, is shown. Each dot represents the median intensity collected from one retina. Numbers of samples analyzed are shown in the graph (data from 4 independent litters). **e** The identities of EdU-labeled cells in the retinas were determined by co-staining of Sox2 and Iba1. The boxed areas in the top row are magnified in subsequent rows. Arrowheads indicate EdU-labeled cell nuclei. (**f**) Quantification of EdU-labeled MG and microglia in the retinas is shown in the graph. Numbers of samples analyzed are shown in the graph (data from 4 independent litters). **g** Composition of EdU-labeled cells in the mouse retinas is displayed in the graph. Error bars denote SEM. P-values were calculated using one-sided Student's t-test (*, $p < 0.05$; ***, $p < 0.005$; ****, $p < 0.001$; n.s., >0.05).

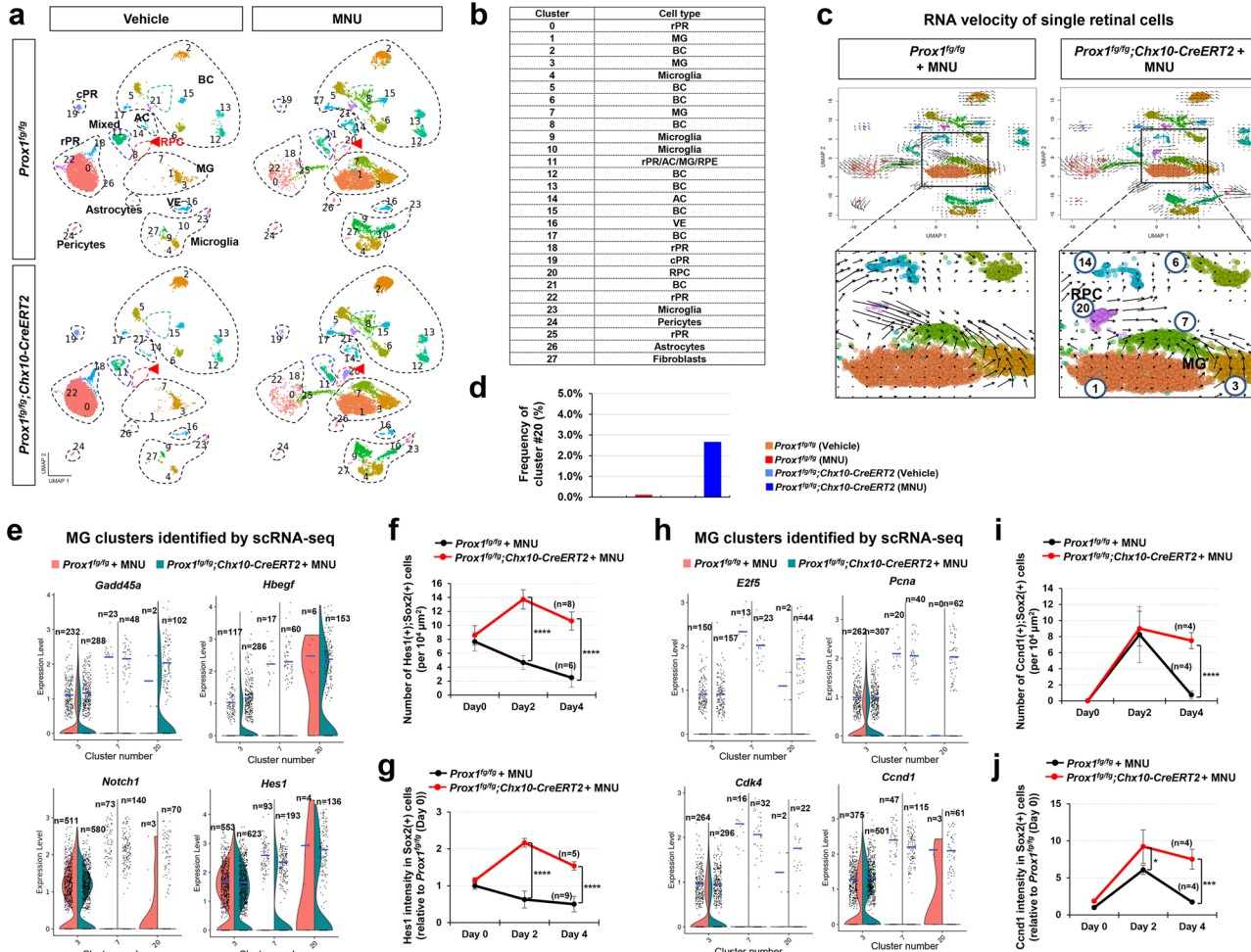

**Fig. 5 | Injury-induced emergence of RPCs in the mouse retina following Prox1 deletion in BCs. a** Single-cell RNA sequencing (scRNA-seq) analysis was performed to examine mRNA expression in cells from *Prox1^{fg/fg}* and *Prox1^{fg/fg};Chx10-CreERT2* mouse retinas before and after MNU injury (see "Methods" for details). Uniform Manifold Approximation and Projection (UMAP) plots show retinal cells clustered by cell type. Red arrowheads indicate cluster #20. **b** The identity of each cell cluster is summarized in the accompanying table. **c** RNA velocity profiles derived from scRNA-seq data for MNU-injured *Prox1^{fg/fg}* and *Prox1^{fg/fg};Chx10-CreERT2* mouse retinas. Magnified views of boxed areas in the top panels are displayed in the bottom. **d** The frequency of cluster #20 in the retinas of the indicated mouse genotypes is shown in the graph. **e** Violin plots depicting the expression levels of *Gadd45a*, *Hbegf*, *Notch1*, and *Hes1* mRNA in the indicated cell clusters identified through scRNA-seq analyses of MNU-injured *Prox1^{fg/fg}* and *Prox1^{fg/fg};Chx10-CreERT2*

mice. Each dot represents gene expression level in individual cells, with the number of dots (n) indicated for each group. Blue horizontal bars indicate mean expression values. **f, g** The distribution of Hes1 in MNU-injured *Prox1^{fg/fg}* and *Prox1^{fg/fg};Chx10-CreERT2* mice mouse retinas was analyzed by immunostaining (images provided in Supplementary Fig. 14a). Graphs show the number of Hes1-expressing MG (**f**) and the intensity of Hes1 in MG (**g**). The number of retinas analyzed is indicated in the graphs. **h** Violin plots illustrate the expression levels of *E2f5*, *Pcna*, *Cdk4*, and *Ccnd1* mRNA in specific cell clusters. **i, j** The distribution of Ccnd1 in retinas was analyzed by immunostaining (images shown in Supplementary Fig. 14b). Graphs display the number of Ccnd1-expressing MG (**i**) and the intensity of Ccnd1 in MG (**j**). Error bars denote SEM. *P*-values were calculated using one-sided Student's t-test (*, $p < 0.05$; ****, $p < 0.001$).

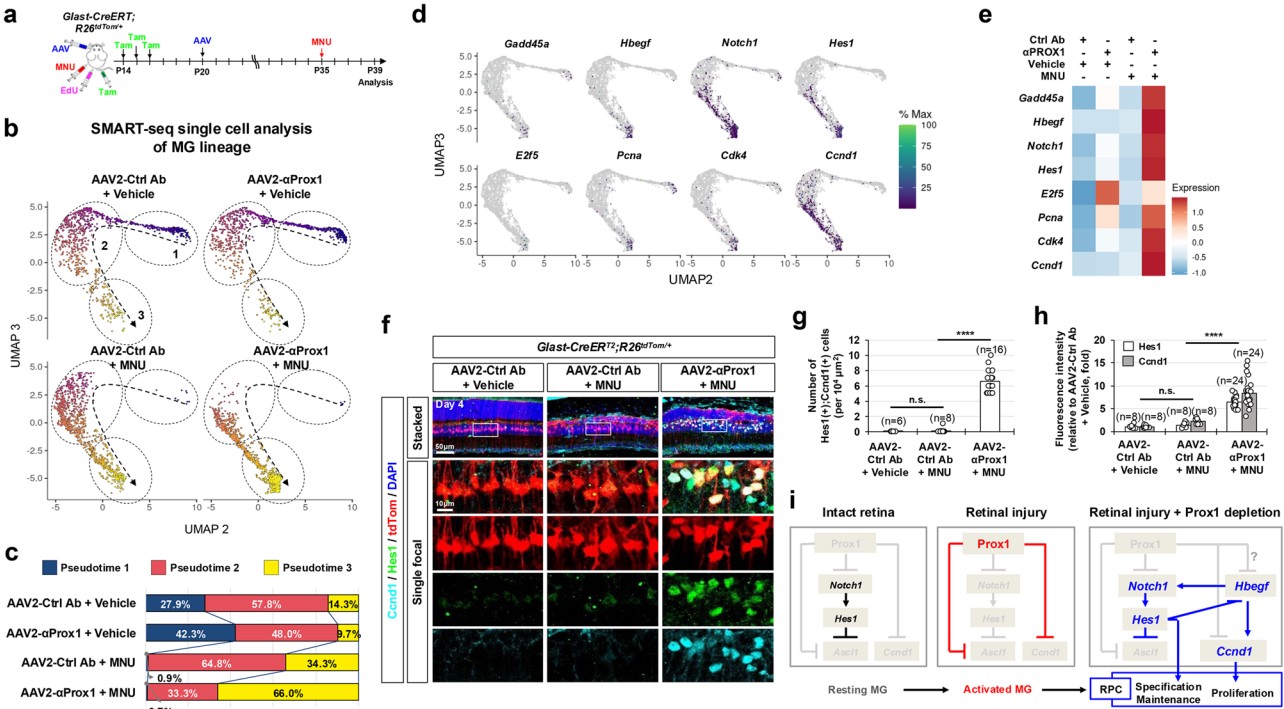

**Fig. 6 | Injury-induced reprogramming of MG into RPCs upon reduced Prox1 transfer. a** Retinas from *Glast-CreERT;R26^{tdTom/+}* mice infected with AAV2-Ctrl Ab or AAV2-αProx1 were injured using MNU. MG lineage cells were then purified by FACS at the indicated time points for SMART-seq analysis (**a–e**) or analyzed by immunostaining for RPC markers (**f–h**). **b** UMAP plots from Monocle 3 analysis of SMART-seq data for FACS-purified tdTom-positive cells in vehicle- or MNU-treated *Glast-CreERT;R26^{tdTom/+}* mouse retinas (details in Methods). Dotted arrows indicate pseudotime progression, with circled areas marking distinct populations by pseudotime intervals. **c** Pseudotime distributions of cell populations for each sample are shown. **d** Pseudotime distributions of cells expressing RPC and proliferative cell marker mRNAs are plotted. **e** Heatmap displaying the mean expression levels of the indicated mRNAs in each sample. **f** Distribution of Ccnd1 and Hes1 in AAV2-infected *Glast-CreERT;R26^{tdTom/+}* mouse retinas was assessed by immunostaining. Magnified views of boxed areas in the top row are shown in the subsequent rows. **g, h** Graphs show the number of Ccnd1 and Hes1 double-positive cells (**g**) and the relative intensity of Ccnd1 and Hes1 in these cells (**h**). Error bars denote SEM. ****, $p < 0.001$; n.s., $> 0.05$ (one-sided Student's t-test). **i** A hypothetical model illustrating the negative regulation of injury-induced MG reprogramming into RPCs by Prox1.

Corresponding to the gain of rPRs, the amplitudes of scotopic electroretinogram (ERG) a-waves were elevated in AAV2-αProx1-infected mice at postnatal day 35 (P35) in comparison to uninfected or *Pde6b^{rd10/rd10}* littermate mice, suggesting that rPR activities remained in the AAV2-αProx1-infected *rd10* mouse retinas (Supplementary Figs. 16a–c). Furthermore, visual acuity of AAV2-αProx1-infected *Pde6b^{rd10/rd10}* mice was significantly higher than that of uninfected or AAV2-Ctrl Ab-infected *Pde6b^{rd10/rd10}* mice (Fig. 7i). However, these effects were not persistent by P60, at which point the visual functions of AAV2-αProx1-infected *Pde6b^{rd10/rd10}* mice were also impaired (Fig. 7i). In contrast, no visual recovery was noted in *Pde6b^{rd10/rd10}* mice infected with AAV2-αProx1 at P60 (Supplementary Fig. 17), indicating that AAV2-anti-Prox1-induced retinal restoration requires early intervention, prior to the completion of retinal degeneration.

**Vision recovery in late-onset retinitis pigmentosa mouse models by blocking Prox1 transfer**

The transient vision recovery observed in AAV2-αProx1-infected *Pde6b^{rd10/rd10}* mice was likely associated with the short lifespan of newly produced PRs, which also carry the *Pde6b^{rd10}* mutation and degenerate in a week (Supplementary Fig. 18). These findings suggest that the regeneration promoting effects of αProx1 might be more effective in retinas where degeneration takes for a longer period. We thus selected *Rp1^{turm64/turm64}* (*Rp1^{m/m}*) mice, which harbor homozygous missense mutations in the *retinitis pigmentosa 1* (*Rp1*) gene and start to lose rPRs from 2 month-old following the shortening of outer segments and gradually lose their vision over 12 months[54] (Fig. 8a, b).

Prox1 protein levels were observed to be higher in MG of *Rp1^{m/m}* mice compared to their littermate *Rp1^{m/+}* mice (Fig. 8a, c), consistent with findings in a RP patient and *Pde6b^{rd10/rd10}* mice (Fig. 7a–e). The increased Prox1 level in MG was mitigated by intravitreal injection of AAV2-αProx1 but not by AAV2-Ctrl Ab (Fig. 8c; Supplementary Fig. 19a–c). Notably, the thickness of the ONL was increased in *Rp1^{m/m}* mice infected with AAV2-αProx1 compared to *Rp1^{m/m}* mice infected with AAV2-Ctrl Ab (Supplementary Fig. 19a, d), despite no significant alteration in apoptotic cell numbers in the mouse retinas (Supplementary Fig. 19a, e). This increase in ONL thickness was associated, at least in part, with the regeneration of rPRs. Significant numbers of EdU-labeled newborn cells in *Rp1^{m/m}* mouse retinas infected with AAV2-αProx1 were Rhodopsin-positive rPRs, whereas EdU incorporation was not observed in the rPRs of littermate *Rp1^{m/m}* mice infected with AAV2-Ctrl Ab (Fig. 8d – f; Supplementary Fig. 19f).

The *Rp1^{m/m}* mice infected with AAV2-αProx1 maintained normal visual acuity, while those receiving AAV2-Ctrl Ab exhibited a gradual decline in vision (Fig. 8g). Furthermore, *Rp1^{m/m}* mice injected with AAV2-αProx1 at 6 months of age showed visual acuity recovery comparable to that of *Rp1^{m/+}* littermate mice within 1 month post-injection (Fig. 8h). However, the visual improvement did not persist beyond 6 months post-treatment (Fig. 8g, h). The limited duration of the effect is likely due to silencing of the EF1α promoter in the AAV2-αProx1 construct, resulting in the loss of αProx1 expression (Supplementary Fig. 20). While the EF1α promoter is commonly used for transient transgene expression in the retina[55], its limitation for sustained in vivo expression has been reported[56–58]. Promoter redesigning is therefore

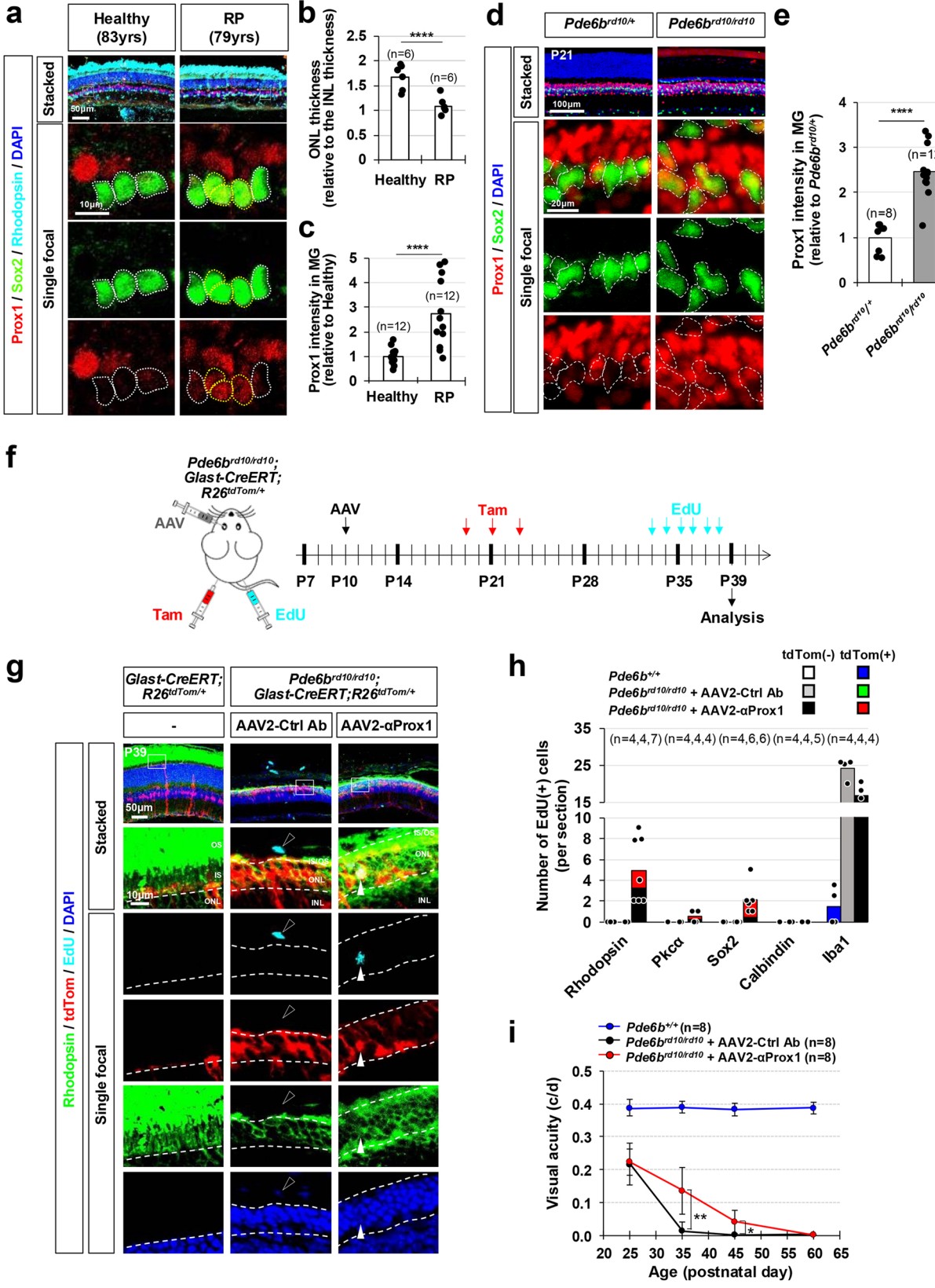

necessary to improve the long-term efficacy of this treatment. In conclusion, blocking PROX1 transfer appears to be a promising therapeutic strategy for late-onset retinal degenerative diseases, where degeneration progresses relatively slower compared to early-onset types.

## Discussion

In the injured zebrafish retina, MG are reprogrammed to RPCs, which reenter the cell cycle to renew themselves and then exit cell cycle to produce retinal neurons[3]. In contrast, in the mouse retina, MG cells rarely convert into RPCs after injury, although they are activated to

**Fig. 7 | Delaying vision loss in early-onset RP model mice via viral gene delivery of anti-Prox1 antibody. a** Retinal sections from a healthy 83-year-old donor eye and a 79-year-old RP patient eye were stained with rabbit anti-Prox1, goat anti-Sox2, and mouse anti-Rhodopsin antibodies. Nuclei of the cells in the retinal sections were visualized by DAPI staining. Sox2-positive MG nuclei are outlined by dotted-lines. **b** The thickness of the ONL in the healthy and RP patient retina was compared to the thickness of the INL in the same section, and the relative values are presented in the graph. Numbers of samples analyzed are shown in the graph (data from 6 independent retinal sections from 2 eyes). ****, $p < 0.001$ (one-sided Student's t-test) (**c**) Relative Prox1 immunofluorescent intensity in MG in the corresponding retina, normalized to Prox1 intensity in BCs within the same image, is shown. Each dot represents the median intensity collected from one retina. Numbers of samples analyzed are shown in the graph (data from 12 independent retinal sections from 2 eyes). *** $p < 0.005$; **** $p < 0.001$ (one-sided Student's t-test). **d** The distribution of Prox1 in the retinas of mice with heterozygous and homozygous $rd10$ mutations in $Pde6b$ gene ($Pde6b^{rd10}$) was investigated by immunostaining. MG cells among Prox1-positive cells were determined by co-staining of Prox1 and Sox2. Boxed areas in top row are magnified in the next three rows. Sox2-positive MG nuclei are outlined by dotted-lines. **e** Prox1 intensity in Sox2-positive MG relative to other retinal neurons in the same image is presented in the graph. Numbers of samples analyzed are shown in the graph (data from 4 independent litters). ** $p < 0.01$; *** $p < 0.005$; **** $p < 0.001$ (one-sided Student's t-test). **f** $Pde6b^{rd10/rd10};Glast-CreERT;R26^{tdTom/+}$ mice were intravitreally injected with AAV2-Ctrl Ab or AAV2-αProx1 at P10, followed by Tam injection to activate the CreERT, resulting in the expression of tdTom Cre reporter in MG. Mice were also injected with EdU daily from P33 to P38 to label cells born during last 6 days before sample collection at P39. **g** The identities of tdTom-expressing MG cell lineage cells in mouse retinas were investigated by co-immunostaining for cell type-specific markers. Immunostaining images are provided in Supplementary Fig. 15f, except for the rPR marker, Rhodopsin. The birth of these cells between P33 and P39 was determined by EdU-labeling. Boxed areas in top row are magnified in the next rows. White arrowhead points EdU-labeled cell rPR nuclei and black arrowhead indicates EdU-labeled cell nucleus in the choroid. **h** Mean numbers of cells expressing corresponding markers in the retinal area are provide in the graph. Numbers of samples analyzed are shown in the graph (data collected from 4 independent litters). **i** Visual acuities of the mice measured at the indicated postnatal days. *, $p < 0.05$; **, $p < 0.01$ (one-sided Student's t-test) Error bars in the graphs represent SEM ($n = 8$; 5 independent litters).

acquire characteristics distinct from those of resting MG[18]. A strong candidate for this missing factor is Ascl1, which has been shown to be capable of triggering cell proliferation upon overexpression in the MG of injured mouse retinas[23,24]. However, the mechanisms that promote *ascl1a* expression in zebrafish MGPCs and suppress Ascl1 in mammalian MG remain unclear.

In *Drosophila* neurogenesis, the expression of fly Ascl1 homologs *scute* and *asense* is suppressed by the fly Prox1 homolog pros[28]. This raises the possibility that the exogenous Prox1 present in the MG of the injured mouse retina could be responsible for suppressing *Ascl1* expression. However, we observed that *Ascl1* expression was not restored in Prox1-depleted MG in *Prox1^{fg/fg};Chx10-CreERT2* mouse retinas or in mouse retinas infected with AAV2-αProx1, whereas the MG regained proliferative capacity (Supplementary Fig. 13b). This absence of *Ascl1* expression in the Prox1-depleted mouse MG or MGPCs is likely related to the upregulation of Notch signaling (Figs. 5e – g, 6d – i), which is known to repress *Ascl1* expression[59].

Thus, the Prox1-depleted MG of the injured mouse retina may be reprogrammed into RPCs through mechanisms independent of *Ascl1* induction, possibly involving other proneural transcription factors. In previous studies, mouse Prox1 has been shown to suppress *Ccnd1* expression in NPCs within the cerebellar external granular layer[49]. Subsequently, this suppression reduces levels of the proneural transcription factor atonal bHLH transcription factor 1 (Atoh1) via the decrease of the Cdk4/Ccnd1 complex, which phosphorylates and stabilizes Atoh1[49]. In RPC populations from the MNU-injured *Prox1^{fg/fg};Chx10-CreERT2* mouse retinas and AAV2-αProx1-infected *Glast-CreERT* mouse retinas, *Ccnd1* levels were found to be elevated (Figs. 5h–j, 6f–h), suggesting a potential increase in Atoh1 in these cells. However, Atoh1 itself was not detectable in the Prox1-depleted mouse retinas. This indicates that the reprogramming of MG into RPCs by Prox1 depletion in these mouse retinas may involve other proneural transcription factors, such as Neurod1, which was significantly expressed in cluster #20 of our single cells RNA-seq data (Supplementary Fig. 13b). Future studies are needed to identify the specific proneural transcription factors that play a critical role in the transition of MG to RPCs.

Among over 300 tdTom-positive retinal cells analyzed, only a single EdU-negative cell was identified in the retinas of *Pde6b^{rd10/rd10};R26^{tdTom/+}* mice, while numerous EdU-labeled cells were observed (Fig. 7h). This single EdU-negative, tdTom-positive cell may represent a neuron either directly transdifferentiated from MG or regenerated from MG prior to the EdU-labeling period. These findings indicate that MG-derived retinal neurons are more likely formed through a regenerative process rather than direct transdifferentiation. However, the possibility that EdU-labeled, newborn MG transdifferentiate into retinal neurons cannot be excluded. To definitively elucidate the mechanism underlying the formation of EdU-labeled, tdTom-positive retinal neurons, real-time tracking of tdTom-positive MG is necessary.

Prox1 is known to be crucial for the differentiation of HCs in the mouse retina during development[31]. However, we observed no evidence of conversion of Prox1-enriched tdTom-positive MG into HCs or other retinal neurons in injured or diseased mouse retinas (Figs. 4g, 7g; Supplementary Fig. 15f). Furthermore, ectopic expression of Prox1 in the MG of *Glast-CreERT;R26^{tdTom/+}* mouse retinas did not result in MG conversion to HCs (Supplementary Fig. 21), suggesting that Prox1 in mouse MG cannot induce spontaneous transdifferentiation to retinal neurons, regardless of its origin. Given that epigenetic modifications are necessary for transdifferentiation of retinal neurons from Ascl1-overexpressing MG[23], these findings imply that altering the epigenetic landscape of MG to resemble that of developing RPCs may be necessary to enable Prox1-mediated conversion of MG into HCs in the adult mouse retina.

Our findings indicate that deleting *Prox1* in BCs was sufficient to reduce Prox1 levels in MG, thereby allowing their reprogramming into RPCs, even though other retinal neurons, such as HCs and ACs, continued to express Prox1 (Fig. 3e, f; Supplementary Fig. 5a–c). Additionally, sequestering extracellular Prox1 using αProx1 was able to lower Prox1 levels in MG, facilitating their conversion into RPCs in the injured mouse retina without affecting endogenous *Prox1* expression in MG (Fig. 4). These observations suggest that a threshold level of Prox1 is necessary to inhibit injury-induced reprogramming of MG into RPCs. Therefore, Prox1 levels in MG may not reach this threshold in these mouse retinas (Figs. 3, 4) and the injured zebrafish retina (Supplementary Fig. 7), allowing MG-to-RPC conversion without the interference by the basal Prox1. However, the precise Prox1 threshold required remains unknown.

Despite the recovery of RPCs in the injured mouse retinas, where Prox1 transfer to MG is suppressed, the number of EdU-labeled cells remains lower than that observed in zebrafish. Specifically, in the MNU-injured *Prox1^{fg/fg};Chx10-CreERT2* mouse retina, we observed an average of ~5 EdU-labeled MG or MGPCs per area, and in the MNU-injured AAV2-anti-Prox1-infected retina, ~3 cells per area, compared to ~28 cells in zebrafish (Figs. 3f, 4f; Supplementary Fig. 7k). These findings suggest that blocking Prox1 alone may be insufficient to fully restore regenerative capacity in the mammalian retina to that of zebrafish retina. Additional events, such as Notch inhibition post-reprogramming and sustained Yap/Taz activation, may be needed to promote robust MGPC proliferation[11,12,16,17]. Our future studies will

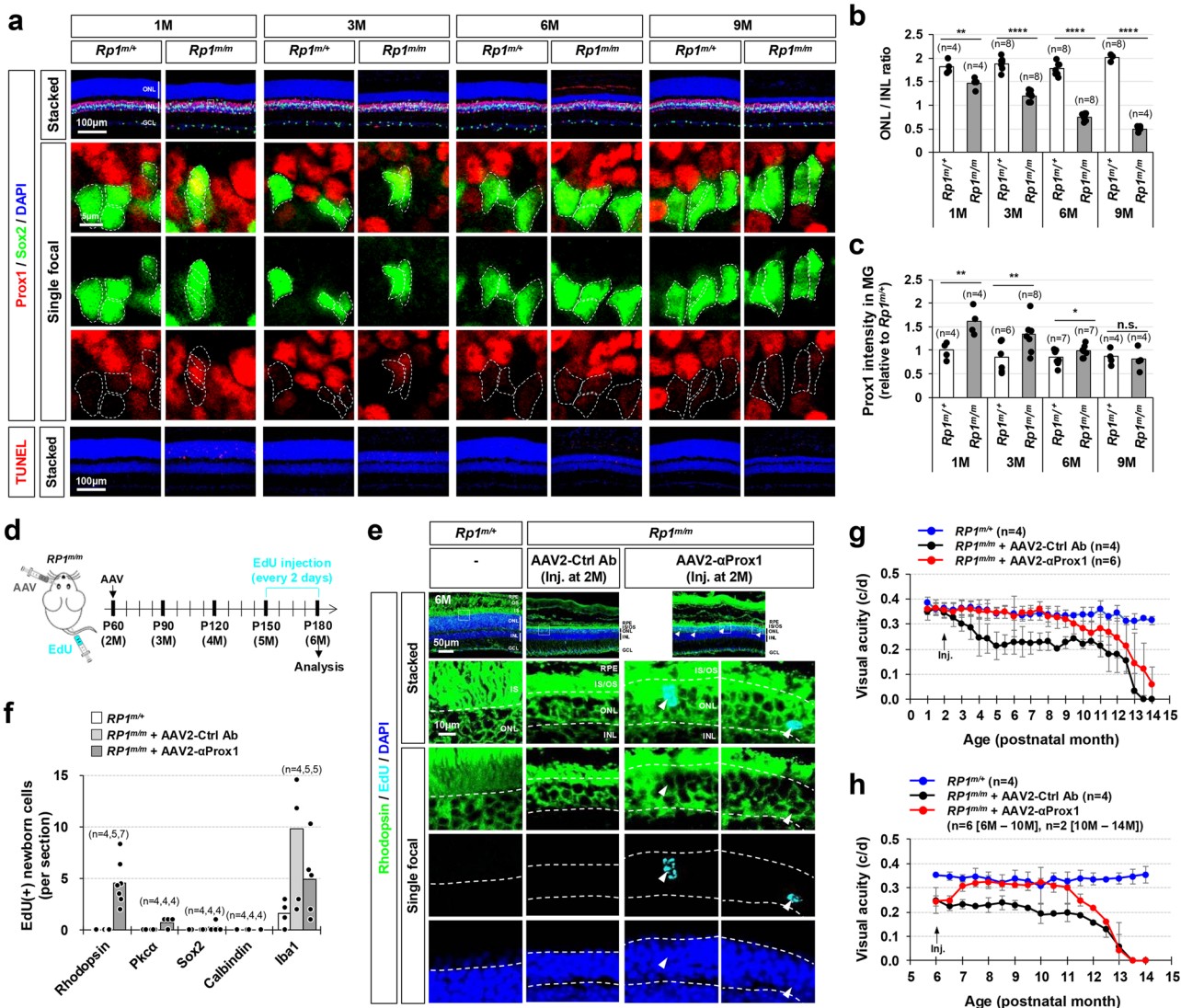

**Fig. 8 | Vision recovery in late-onset RP model mice by viral gene delivery of anti-Prox1 antibody.** (**a**) The distribution of Prox1 in the retinas of mice with heterozygous and homozygous *tvrm64* mutation of *Rp1* gene (*Rp1^m*) was investigated by immunostaining. MG cells among Prox1-positive cells were determined by co-staining of Prox1 and Sox2. Boxed areas in top row are magnified in the next three rows. Sox2-positive MG nuclei are outlined by dotted-lines. (**b**) The thicknesses of the ONL and INL of the retinal sections were measured and relative thickness of the ONL compared to the INL of the same retinal sections is presented in the graph. **, *p* < 0.01; ****, *p* < 0.001 (One-sided Student's t-test). (**c**) Relative Prox1 immunofluorescent intensity in MG in the corresponding retina, normalized to Prox1 intensity in BCs within the same image, is shown. Each dot represents the median intensity collected from one retina (data from 3 independent litters). **, *p* < 0.01; ***, *p* < 0.005; ****, *p* < 0.001 (One-sided Student's t-test). (**d**) *Rp1^m/m* mice

were intravitreally injected with AAV2-Ctrl Ab or AAV2-αProx1 at P60, and then with EdU every 2 days for one month before sample collection to label the cells born around the injection time points. (**e**) The identities of these EdU-labeled cells in mouse retinas were investigated by co-immunostaining for cell type-specific markers. Immunostaining images are provided in Supplementary Fig. 19f, except for the rPR marker Rhodopsin. Boxed areas in top row are magnified in the next rows. Arrowheads, EdU-labeled cells containing Rhodopsin. (**f**) Number of cells expressing corresponding markers (data collected from 2 independent litters). (**g**) Visual acuities of the mice infected with AAV at 2 months of age (2 M) and measured at the indicated postnatal months. Error bars in the graphs represent SEM (3 independent litters). (**h**) Visual acuities of the mice infected with AAV at 6 M and measured at the indicated postnatal months. Error bars in the graphs represent SEM (3 independent litters).

focus on exploring these adjunctive triggers to enhance the regenerative efficiency of the anti-Prox1 approach.

MG not only extend processes to the inner and outer retinal surfaces to form barriers, but also branch into the outer and inner plexiform layers (OPL and IPL) to modulate synaptic transmission among retinal neurons[60]. Thus, MG may capture Prox1 secreted from BCs into the OPL and IPL, where BCs form pre- and post-synaptic connections, respectively (Figs. 3a, 4a, model diagram). Supporting this, we detected a significant decrease in Prox1 levels specifically in MG in the retinas of *Prox1^fg/fg;Chx10-CreERT2* mice (Supplementary Fig. 5a, b). However, we found no changes in the Prox1 levels in ACs or HCs, which form synapses with BCs in the OPL and IPL, respectively, in these mouse retinas

(Supplementary Fig. 5a,b). This suggests that extracellular Prox1 preferentially transfers to MG over ACs and HCs in the mouse retina.

The selective transfer of transcription factors over short or long distances has been noted in other contexts. For example, orthodenticle homeobox 2 (Otx2) can move short distances across synapses between PRs and BCs[61] and long distance from the choroid plexus to cortical parvalbumin (PV) interneurons[62]. In the latter case, Otx2 selectively binds to chondroitin–4-sulfate, enriched in the perineural net surrounding PV interneurons, enhancing Otx2 uptake specificity[63,64]. A similar process may also be at work in MG, which may trap Prox1 by expressing specific types of heparan or chondroitin sulfate proteoglycans that are absent in other retinal neurons.

Interestingly, enzymes involved in introducing chondroitin sulfate (*Chst1* and *Chst5*) and heparan sulfate (*Ext1, Ext2, Extl1, Extl2, Extl3, Ndst1, Ndst2, Ndst3,* and *Ndst4*) are commonly upregulated in MG lineage 6–12 h after NMDA injury in mice[18]. In contrast, the levels of these genes do not change in the injured zebrafish retina[18], where Prox1 levels remain unchanged in MG after injury (Fig. 1b, c). However, due to the selective, yet redundant roles of these enzymes, future studies will be necessary to identify which specific enzymes contribute to the selective transfer of Prox1 into MG.

Although visual acuity was fully restored in *Rp1^{m/m}* mice infected with AAV2-αProx1 (Fig. 8g, h), the ONL thickness and scotopic ERG responses were not completely normalized (Fig. 8a, b; Supplementary Fig. 22a–c). These results suggest that retinal regeneration induced by AAV2-αProx1 infection was insufficient to fully restore the rPRs in *Rp1^{m/m}* mice. Instead, these mice exhibited an increase in cPR numbers and photopic ERG responses, which are attributed to the function of cPRs (Supplementary Fig. 22d–f; Supplementary Fig. 23). In patients with RP, cPR degeneration typically follows rPR degeneration, ultimately leading to vision loss, as cPR survival relies on factors produced by rPRs. One such pathway involves the rod-derived cone viability factor (RdCVF), expressed in rPRs to support cPR survival[65]. Thus, the partial recovery of Rhodopsin-expressing rPRs in AAV2-αProx1-infected *Rp1^{m/m}* mice may be sufficient to provide critical protective factors to cPRs, promoting their survival by exceeding a necessary threshold concentration.

## Methods

### Experimental model and subject details
**Human samples.** Human donor eyes (bank tissue ID are 22–1465 [RP patient, 79 years-old] and 22–1587 [healthy donor, 83 years-old]) were collected in collaboration with the Lionizers Gift of Sight Eye Bank according to a standardized protocol of the bank. Briefly, eyes were procured within a 12-h post-mortem interval. To determine the ocular phenotype relative to disease and healthy aging, ophthalmoscopic analysis was performed by a team of retinal specialists and ophthalmologists at the Lionizers Gift of Sight Research Team. This diagnosis was then compared to medical records.

The isolated eyes were fixed in phosphate buffered saline (PBS) containing 4% paraformaldehyde (PFA) and delivered to the laboratory in a container filled with icepacks within 36 h. The fixed eyes were dissected to quadrants for subsequent cryopreservation in OCT (Tissue-Tak). The frozen eye sections were used for the immunohistochemical analyzes described in the following. The donor concents for the hyman eye tissues were collected by the Lionizers Gift of Sight Eye Bank. All procedures were done according to the protocol approved by KAIST Institutional Review Board (IRB-22-551).

**Animals.** *Prox1^{tm1.1Fuma}* (*Prox1^{fg}*)[30], *Dkk3^{tm1Tfur}* (*Dkk3-Cre*)[41], *Tg(Chx10-cre/ERT2)G7Tfur* (*Chx10-CreERT2*), *Tg(Slc1a3-cre/ERT)1Nat/J* (*Glast-CreERT*)[40], *Tg(Crx-cre/ERT2)1Tfur* (*Crx-CreERT2*)[38], *B6.CXB1-Pde6b^{rd10}/J* (*Pde6b^{rd10}*), and *C57BL/6J-Rp1^{turm64}/PjnMmjax* (*Rp1^m*)[54] mice were maintained on a 12-hour light/dark cycle and had access to standard mouse chow and water ad libitum in a specific pathogen-free mouse facility of KAIST Laboratory Animal Resource Center. The facility maintains macroenvironmental temperature and humidity ranges of 22 °C to 24 °C and 45% to 55%, respectively. For intravitreal injections, mice were anesthetized with isoflurane (1.5% induction and 1.0% maintenance). Samples loaded into a Hamilton syringe equipped with a blunt 33-gauge needle were then injected into the intravitreal space of the mouse eye. Euthanasia of mice was humanely performed using $CO_2$ inhalation in accordance with institutional guidelines and ethical regulations. All of the animals were handled according to approved institutional animal care and use committee (IACUC) protocols (KA13-130 and KA2019-14) of Korea Advanced Institute of Science and Technology (KAIST).

AB (wild-type) and *Tg(gfap:EGFP)* zebrafish (*Danio rerio*) strains were maintained at 28.5 ± 1 °C with a 14/10-hour light/dark cycle. To injure the fish retina using MNU, adult zebrafish (6-12 months of age) were incubated in fresh media containing MNU (150 mg/l) and 10 mM sodium phosphate (pH 6.3) for 1 h and then washed with system water[66]. The MNU-treated zebrafish were maintained under standard husbandry conditions. To label proliferating cells, the zebrafish were anesthetized with 0.04% Tricaine (ethyl 3-aminobenzoate methanesulfonate salt, #A5040, Sigma, USA) in system water and given *i.p.* injections of 20 µl of 50 mg/ml EdU (dissolved in PBS) at the specified times. All zebrafish care and husbandry were performed in compliance with the guidelines of the Korea Research Institute of Bioscience and Biotechnology (KRIBB) IACUC (KRIBB-AEC-20235).

### Experimental methods
**Immunohistochemistry.** Proteins in the retinal sections and culture slides were detected by immunostaining, respectively, as described previously[61]. In brief, sections (20 µm) of frozen zebrafish and mouse eyes were incubated in a blocking solution (PBS including 10% normal donkey serum and 0.1% Triton X-100) at room temperature for 2 h. The sections were further incubated in blocking solution (without Triton X-100) containing primary antibodies at 4 °C for 16 h, and then with fluorophore-conjugated secondary antibodies that recognize the primary antibodies. The information on antibodies used in this study is provided in the Supplementary Table 1. Fluorescent images were then obtained using Olympus FV1000 and FV3000 confocal microscopes.

**Cell culture and transfection.** HeLa cells were maintained in Dulbecco's Modified Eagle Medium (DMEM) containing 10% fetal bovine serum (FBS) and transfected using GenJet Plus DNA in vitro transfection reagent (SignaGen) according to the manufacturer's instructions. Sf9 cells were maintained in Grace's insect medium containing 10% FBS, 1% poloxamer 188 solution, and 0.1% gentamicin. Sf9 cells were transfected using Cellfectin II reagent (Gibco) according to the manufacturer's instructions.

**Quantification of immunostaining signals.** Single focal confocal microscopic images were converted to grayscale and utilized for quantifying fluorescent signal intensity. Subsequently, the cell images were adapted to facilitate nucleus segregation analysis. Prox1 signals in MG were captured within Sox2+ areas. For acquiring Prox1 signals in BCs and ACs, the outermost 2–3 layers of the inner nuclear layer (INL) were delineated as the BC region, while the innermost single layer of the INL was designated as the AC region. These regions were delineated using the lasso tool in Photoshop and then exported. GFP images were binarized using the Otsu algorithm in Python, and the nuclei on DAPI images containing >50% of binary GFP in nucleus areas were selected for quantifying fluorescence intensity.

Prox1 images and the modified target cell images were imported into Python. Nucleus segregation of both Prox1 and target cell images was conducted using the StarDist algorithm[67,68]. The Prox1 image was normalized by the mean value of Prox1 fluorescent intensity in segregated Prox1+ nuclei. Segregated target nuclei with >200 pixel counts were utilized to quantify Prox1 fluorescent intensity. To minimize spillover Prox1 fluorescence from neighboring Prox1+ cells, overlapping Prox1+ nuclei with the selected single target nucleus were identified. If the overlapping Prox1+ nucleus was not a subset of the target nucleus, the Sørensen–Dice coefficient (SDC) between the target nucleus and each overlapping Prox1+ nucleus was calculated to define overlapping regions with an SDC < 0.5 as 'spill-over' regions. The target nucleus region was then adjusted by excluding these spill-over regions

$$SDC_i = \frac{2 * (A_i \cap B)}{(A_i \cup B)}, \text{if } A_i \nsubseteq B$$

Where A is i-th neighbor Prox1+ nucleus region and B is a target nucleus region.

The pixel intensity matrix of the corrected target nucleus region was then extracted from normalized Prox1 images. The median value of the pixel intensity matrix was computed and represented as the Prox1 intensity value of a single target nucleus. After obtaining median values from all target nuclei, the mean of these median values was considered as the Prox1 intensity value of one single-focal image, which was used for subsequent analysis and plotting.

To quantify GFP signal intensity, segregated target nuclei with >200 pixel counts were utilized. Histogram equalization was applied to each GFP image using the 'cv2.equalizeHist()' function for comparison with *Prox1fg/+;Dkk3-Cre* mouse retinal images. GFP intensity was measured using the same method as for Prox1 intensity measurement, with the exception of spill-over correction

**RNAscope in situ hybridization.** RNAscope in situ hybridization was conducted to detect mRNA expression in mouse retinal sections following the manufacturer's instructions. For co-staining of Prox1 mRNA and Sox9 protein, the RNA-Protein Co-Detection Ancillary Kit (Cat. #323180) and a Sox9 antibody were employed. The Multiplex Fluorescent V2 Assay utilized to probe Prox1 (Probe- Mm-Prox1-C1, Cat. # 488591, ACDbio), while Sox2 protein was detected by immunostaining. Subsequently, the fluorescence signals from the Prox1 probe and Cy3-labeled donkey anti-rabbit secondary antibody were visualized using confocal microscopy

**Quantification of Prox1 RNAscope signals.** Single-focal target cell images and RNAscope images were imported on Python and nucleus segregation was performed using Stardist software. *Prox1* RNAscope signals inside segregated target nuclei (>200 pixel-counts) were exported as gray-scaled images for quantification. Imported *Prox1* RNAscope images on ImageJ were converted to binary image by using 'Threshold' tool (minimum threshold value = 1, maximum threshold value = 255, and activated- 'dark background' option). The particle size and number of binary *Prox1* RNAscope signals were measured through 'Analyze Particles' tool with activated- 'Include holes' option. The particle size and number were divided by the number of segregated nuclei.

**Recombinant Prox1 protein production and intravitreal injection.** The PROX1 cDNA was subcloned into pFASTBAC1 to express PROX1 with an N-terminal FLAG-tag in Sf9 cells by the baculovirus expression system (Invitrogen). The PROX1-FLAG protein was then purified on M2 agarose (Sigma), which captures FLAG-tagged proteins as described previously[69]. FLAG peptide (Sigma, cat#F4799), with the sequence DYKDDDDK, was prepared by diluting it in sterilized PBS to the same molar concentration as the PROX1-FLAG protein solution. The purified proteins were diluted in sterilized PBS and 1 μl of the protein/PBS solution loaded into a Hamilton syringe equipped with a blunt 33-gauge needle for injection into the intravitreal space of the mouse and zebrafish eyes. Contralateral eyes received a sham injection of PBS alone and served as controls.

**Subretinal DNA electroporation.** Electroporation experiments were performed, as previously described[70]. In brief, ~0.5 μl (2.5 μg) DNA solution mixed with fast green dye was injected into the subretinal space of P4 mouse retinas, and square electric pulses were applied (100 V; five 50-ms pulses at 950-ms intervals).

**Isolation of αProx1 clone.** An antibody with affinity for Prox1 was screened from a phage display library of chicken immunoglobulin genes by YntoAb Inc. A clone (1A11) that exhibited strong affinity to the antigen was selected for further in vitro and in vivo analyzes. The immunoglobulin gene clone was prepared as a single-chain fragment variable (scFv) and fused with hemagglutinin (HA) and 6×His tags (6XHis-HA) for

detection purposes during phage display screening. To serve as a control, an scFv without affinity to Prox1 was also identified from the same phage library. This control scFv was prepared following the same procedure as the αProx1 scFv and was used to generate a control antibody construct (AAV2-Ctrl Ab) for experimental comparisons.

**AAV preparation.** The scFv construct targeting Prox1 was fused with the signal sequence of human interleukin-2 (IL2) and introduced into the genome of an adeno-associated virus (AAV) under the control of an EF1α promoter. AAV2-αProx1 was produced in HEK293T cells using a triple transfection method with plasmids encoding the AAV2/2 capsid, the scFv transgene, and a helper plasmid. Viral particles were purified using iodixanol gradient ultracentrifugation, and their titer was confirmed by quantitative PCR targeting the inverted terminal repeat (ITR) sequences. AAV2-Ctrl Ab, expressing an scFv with no affinity to Prox1, was produced and purified using the same method. For intravitreal injection, purified AAV2-Ctrl Ab and AAV2-αProx1 were diluted in sterilized PBS. A 1 μl solution containing $5 \times 10^9$ genome copies was injected into the intravitreal space of the mouse eye.

**Single cell RNA analysis: cDNA library synthesis.** Freshly prepared samples were resuspended in the resuspension buffer (PBS containing 0.04% BSA and 0.5U/ml RNase inhibitor). Dissociated cells were then filtered through a 50-ml cell filter. Cells ( ~ 10,000) were loaded into a 10x Genomics Chromium Next GEM Single Cell system (10x Genomics). Single cell cDNA libraries were prepared using the Chromium Next GEM Single Cell 3′ Kit according to the manufacturer's instructions. Synthesized cDNA libraries were pooled and sequenced on Illumina pair ended sequencing system.

**Single cell RNA analysis: quality control and visualization of scRNA-seq data.** Raw sequencing files were converted to FASTQ files by the cellrnager mkfastq software, and then to gene-cell matrices by the cellranger count command. Gene-cell matrices were analyzed by the Seurat package to visualize scRNA-seq data[71]. Cells that have >20% mitochondrial counts or less than 100 nFeature value were excluded, and the filtered cells was processed through Seurat integration pipeline to cluster them on UMAP using t-distributed stochastic neighbor embedding (tSNE)[72]. The clusters visualized on UAMP were annotated by cell type-specific markers. Retinal cells expressing specific genes were quantified on RNA slot. Mean cross bar for vlnplot was processed by stat_summary and mean crossbar data was merged with vlnplot data on Adobe Photoshop 2021.

**Single cell RNA analysis: RNA velocity analysis.** Spliced/unspliced dataset was calculated by velocyto package to obtain loom files. The loom files were then loaded on R studio through the velocyto.R and SeuratWrappers R libraries to convert into Seurat class. Spliced slot was normalized by SCTransfrom to pull the count data and UMAP reduction was proceeded. Velocity value was calculated by Runvelocity and was projected onto embeddings coordinates of integration-seurat-object.

**SMART-seq: Single cell cDNA synthesis and sequencing library generation.** Single-cell RNA sequencing (scRNA-seq) libraries were prepared using the Smart-seq2 protocol with few modifications[73]. Single-cells were sorted using a BD FACSAria Fusion (BD Biosciences) into 96 well PCR plates (Thermo Scientific) containing 2 μl of lysis buffer (0.1% Triton X-100, 1U/μl RNase Inhibitor (Enzynomics), 0.25 μM oligo-dT30VN primer) and stored at -80°C. The plates were thawed on ice. Revers transcription (20U/μl Maxima H minus transcriptase, 1 M Betaine, 5 mM MgCl2, 1 μM template switching oligo, additional 0.8U/μl RNase Inhibitor), template switching reaction and PCR pre-amplification (KAPA HiFi HotStart (Roche), 18 cycles) were performed according to the protocol. The PCR products in each well was cleaned up using 0.6× SPRI beads (2% Sera-Mag Speed Beads (Cytiva), 1 M NaCl, 10 mM Tris-HCl pH

8.0, 1 mM EDTA, 0.01% NP40, 0.05% Sodium Azide, 22% w/v PEG 8000). The quality of cDNA libraries was assessed by quantitative PCR with a primer pair of GAPDH–a housekeeping gene (forward: 5′-GTCGTGGAGTCTACTGGTGTCTTCAC-3′; reverse: 5′-GTTGTCA-TATTTCTCGTGGTTCACACCC-3′). 50–100 pg of each cDNA library were used to generate the Illumina sequencing library using EZ-Tera XT DNA library preparation kits (Enzynomics). After the final PCR amplification, samples were pooled and cleaned by MinElute PCR purification kit (Qiagen). Using 0.3× and 0.6× SPRI beads, fragments of ~200 bp length were selected as library and confirmed by High Sensitivity DNA ScreenTape Analysis (Agilent). Pooled and size-selected sequencing libraries were sequenced on an Illumina NextSeq 550 instrument using a High Output kit (Illumina) with 38-bp paired-end reads setting.

**SMART-seq: scRNA-Seq data analysis.** Sequencing reads from single-cell RNA-Seq libraries were aligned to the mouse reference genome (version mm10 from the UCSC) using RSEM (version 1.3.1) with STAR (version 2.7.9a) aligner with default parameters for paired-end reads[74]. The matrix of raw counts was transformed into Seurat object (version 4.0.2) for the downstream analysis. SMART-seq data was then analyzed on Seurat[75,76] and monocle3[77,78]. Each raw count seurat objects were normalized by SCTransform and were integrated. Integrated Seurat object was converted to cell data set object compatible with monocle3 by as.cell_data_set(). Converted integrated-object was performed by common workflow (pre-processing, dimensionality reduction and clustering cells). To analyze trajectory analysis, the cell farthest from AAV2-αProx1 + MNU sample on UMAP was specified as the root node by order_cells(). Gene expression figure by pseudotime was plotted by plot_cells() command. After cluster number in monocle object was inserted into integrated Seurat object, heatmap was plotted by DoHeatmap() command.

**Visual acuity test by OptoMotry.** Mouse visual acuity was measured by the OptoMotry system (Cerebral Mechanics) as described previously[79]. In brief, mice were adapted to ambient light for 30 min and placed on a platform 60 cm above the floor and surrounded by computer monitors displaying black and white vertical stripe patterns. The mice stopping movement to begin tracking the stripe movements with head-turn was counted as a successful visual detection. The detection thresholds were then obtained from the OptoMotry software.

**Electroretinogram (ERG).** Mice were kept in the dark for 12 h before scotopic ERG recording and anesthetized with isoflurane. A gold-plated objective lens was placed on the cornea and silver-embedded needle electrodes placed at the forehead and tail after the pupils were dilated by 0.5% tropicamide. The ERG recordings were performed using a Micron IV retinal imaging microscope (Phoenix Research Labs) and analyzed by Labscribe ERG software according to the manufacturer's instructions. A digital bandpass filter ranging from 0.3 to 1000 Hz and stimulus ranging from −2.2 to 2.2 log(cd·s m$^{-2}$) were used to obtain scotopic ERG waves, and a filter ranging from 2 to 200 Hz and stimulus ranging from 0.4 to 2.2 log(cd·s m$^{-2}$) with 1.3 log(cd·s m$^{-2}$) background were used to obtain photopic ERG waves.

**Statistics and reproducibility**
**Statistical analysis.** Statistical tests were performed in Prism Software (GraphPad; v5.0). All data from statistical analyzes are presented as the mean ± standard error (STE). Comparisons between two groups were made by unpaired Student's t-test, and the differences among multiple groups were determined by analysis of variance (ANOVA) with Tukey's post-test. $P$-values < 0.05 were considered significant.

**Abbreviations used in this paper**
Müller glia, MG; prospero-related homeobox 1, Prox1; central nervous system, CNS; retinal progenitor cell, RPC; MG-derived retinal progenitor cell, MGPC; rod photoreceptor, rPR; retinal ganglion cell, RGC; retinitis pigmentosa, RP; hairy and enhancer of split-1, Hes1; heparin-binding epidermal growth factor, Hbegf; yes-associated protein, Yap; WW domain-containing transcription regulator 1, TAZ; achaete-scute family bHLH transcription factor 1, Ascl1; horizontal cell, HC; bipolar cell, BC, amacrine cell, AC; homeodomain protein, HP; N-methyl-N-nitrosourea, MNU; N-methyl-D-aspartate, NMDA; diphtheria toxin A, DTA; tamoxifen, Tam; tdTomato, tdTom; fluorescence-activated cell sorting, FACS; glutamate aspartate transporter, Glast; glial fibrillary acidic protein, gfap; adeno-associated virus, AAV; anti-Prox1 antibody, αProx1; control, Ctrl; single-cell RNA sequencing, scRNA-seq; Switching Mechanism At the 5′ end of RNA Template, SMART; retinal dystrophy-10, rd10; phosphodiesterase 6b gene, Pde6b; cone photoreceptor, cPR; outer nuclear layer, ONL; rod bipolar cell, rBC; electroretinogram, ERG; retinitis pigmentosa 1, Rp1; atonal bHLH transcription factor 1, Atoh1; outer plexiform layer, OPL; inner plexiform layer, IPL.

## Reporting summary

Further information on research design is available in the Nature Portfolio Reporting Summary linked to this article.

## Data availability

All data generated or analyzed during this study are included in this published article and Supplementary Information. Source data are provided as a Source Data file. The scRNA-seq dataset generated in this study has been deposited in Gene Expression Omnibus (GEO) under the accession numbers GSE290239 and GSE290470. We also deposited the dataset the Korea BioData Station under the under accession numbers KAP240958 and KAP240959. Other data that support the findings of this study are available from the corresponding author upon reasonable request. Source data are provided with this paper.

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

## Acknowledgements

We are grateful to Drs. Takahisa Furukawa and Jeremy Nathans for generously providing *Dkk3-Cre, Chx10-CreERT2, Crx-CreERT2* and *Glast-CreERT* mice, which were instrumental for our research. We also extend our sincere appreciation to those who donated their eyes for this study, enabling us to further our understanding in this field. This work was supported by the National Research Foundation of Korea (NRF) grants (RS-2018R1A5A1024261 [JWK] and RS-2022-NR070536 [JWK]); Korea Drug Development Fund (RS-2023-00258166 [JWK]); Korean ALPA-H Innitiative (RS-2024-00512384 [JWK]); and Tech Incubator Program for Startup Korea (RS-2023-0258330 [EJL]).

## Author contributions

E.J.L., M.K., and J.W.K. wrote the manuscript; E.J.L., M.K., S.P., J.H.S., H.J.C., J.A.P., K.P., D.L., H.J., and J.H.K. designed and performed the experiments and analyzed the data; F.M. provided an experimental model; S.Y.K., J.K., H.Y., J.S.L., and J.W.K. supervised the project.

## Competing interests

E.J.L, S.P., J.H.S., J.A.P., and J.W.K. are co-founders or employees of Celliaz Inc., which develops anti-PROX1 therapeutics for retinal diseases. The remaining authors declare no competing interests.
