## [Transparent Peer Review file · Nature Communications]

Restoration of retinal regenerative potential of Müller glia by disrupting intercellular Prox1 transfer

Corresponding Author: Professor Jin Woo Kim

Version 0:

Reviewer comments:

Reviewer #1

(Remarks to the Author)

Summary

The authors set out to restore lost photoreceptors and visual acuity after retinal injury and degeneration by inducing mouse Müller glia (MG) proliferation and neurogenesis via the disruption of PROX1 accumulation. There is evidence that PROX1 suppresses neurogenic transcription factor *asc* and cell cycling factor cyclin E in *drosophila*. The authors first establish that PROX1 protein accumulates in the mouse MG after NMDA and MNU induced retinal injuries via immunostaining; this was compared to the pro-proliferative fish MG which did not demonstrate such accumulation. Despite the increase in PROX1 protein, the authors failed to find a corresponding increase in mRNA expression in the mouse MG through RNA in situ or RT-qPCR. After knocking out the endogenous *Prox1* locus specifically in the MG by *Glast-CreERT* based tamoxifen induced recombination, the authors note that fluorescent markers for endogenous *Prox1* locus expression failed to appear during NMDA and MNU damage, signifying the successful knockout of endogenous *Prox1*. However, PROX1 protein still accumulated in the MG despite a failure to transcribe the endogenous *Prox1* locus. The authors concluded that PROX1 accumulation in the mouse MG after injury must have an exogenous source.

Cell to cell transport of PROX1 is possible, and bipolar cells appeared to have the highest amount of *Prox1* expression; thus, the authors knocked out *Prox1* specifically in bipolar cells through *Chx10-CreERT2* inducible recombination. This finally led to a decrease in PROX1 accumulation in the injured MG, which also had a corresponding increase in proliferating MG.

These results were reversed through an intravitreal injection of the PROX1 protein, showing that they were indeed due to the relief of PROX1 accumulation. Exogenous PROX1 delivery to the zebrafish retina even decreased *Ascl1a* expression and the regenerative potential of the zebrafish MG. Using an in vitro verified PROX1 antibody, the authors then sequestered extracellular PROX1 through AAV2 gene delivery. A similar increase in MG proliferation was induced. The transcriptome of the PROX1-mediated MG was then quantified using scRNAseq, which showed a unique population of MGs only in the case of exogenous PROX1 removal via both AAV-anti-PROX1 and *CHx10-CreERT2*. Notch effector genes such as *Hes1* as well as cell cycle genes such as *Ccnd1* and *Ccne1* were shown to be affected by PROX1 reduction. The newly generated proliferative MG appear more retinal progenitor like.

Finally, the authors set out to see the effects of PROX1 reduction in the degenerating retina. First, human retinas with retinitis pigmentosa were found to have elevated PROX1 in the MG as in the injured mouse retina. Then, in the *rd10* mice, PROX1 reduction through anti-PROX1 antibody not only maintained a thicker outer nuclear layer compared to *rd10* controls, but visual function was restored both through ERG readings as well as optomotor experiments. The proliferating MG appear to differentiate into photoreceptors and bipolar cells. The restoration of visual function was short-lived; thus, the authors used a slower degenerating mouse model of *Rp1* mice. The same maintenance of outer nuclear layer as well as visual function of ERG were observed; this effect now lasted months after anti-PROX1 treatment. The authors contend that PROX1 accumulation in mammalian MG in the injured or degenerating retina suppresses pro-proliferative factors such as *Ascl1*, *Ccnd1*, *Notch1*, and *Hes1*. In turn, the elimination of PROX1 accumulation led to the dedifferentiation of MG into retinal progenitor cells, proliferation, and restoration of lost neurons in the disrupted retina.

Major Concerns

1. Overall: The impact of the PROX1 induced regeneration, the crux of the paper, appears to be relatively small. In cases of non-relative measures, less than 5 EdU cells per $10^4 \mu\text{m}^2$ were observed after PROX1 disruption. Yet this led to significant ONL preservation as well as vision restoration. This could be due to the arbitrary amount and timing of the EdU injections failing to capture the real amount of proliferating MG (although this could be partially resolved by the quantification of

coexpression of EdU-Tdt+ cells with neuronal markers). More measured EdU injections should be done to capture more proliferating cells. The prospect of transdifferentiation and neuropreservation are also not really explored in this study. Even though ERG and visual acuity measurements in the retinal degenerative mice are promising, more work is needed to tease out the connection between PROX1 function and these measures.

2. Figure 1 and figure ED1 are supposed to compare different injury paradigms and PROX1 accumulation. Please include experimental timeline for these figures and all subsequent figures where appropriate. Fig. 1a and 1b show days post MNU injury while 1d-g neglect to mention which of these MNU time points were used. Also, comparison of cell death between the different injury paradigms is important in order to correlate their cell death with MG activation/proliferation.

3. A couple of inducible Cre mouse models such as Glast-CreERT2 and Chx10-CreERT2 were used in the paper. Although these mouse lines were characterized in previous papers, much of the authors' results rely on their expression specificity after crossing them with new lines; thus, expression specificity and recombination efficiency must be established in the paper. This leads to problems such as questioning the bipolar cell specific nature of the Chx10 mouse. According to the Blackshaw data used in the paper, Chx10 is not bipolar specific in the mouse retina. In fact, the mouse MG appear to express Chx10. The exogenous nature of PROX1 then becomes muddled.

4. The paper relies on relative measures throughout. PROX1 accumulation is measured through fluorescent amount/intensity relative to neurons, even though PROX1 expression in neurons across different models and paradigms are not guaranteed to be comparable. Dkk3-Cre is used as a control comparison in figure 1h and ED3b even though such a mouse line is never mentioned in the body or the reason for the comparison given. Retinal degeneration is measured through ONL to INL thickness comparison in figure 4, 5, and on. Even though relative measures are important and sometimes the most telling measures, absolute measures are also sometimes required to give a better understanding of the magnitude of the changes observed. This leads to the next major concern.

5. Page 7 bottom paragraph: The sequestration of extracellular PROX1 using anti-PROX1 antibody in figure 2 leads to the question of whether such measures (including Chx10 based recombination) actually eliminated the transfer of PROX1 from exogenous sources into MG in vivo. It is stated that anti-Prox1 is acting in the extracellular space to inhibit Prox1 uptake by Muller glia. Please provide data indicating cellular expression of anti-Prox1 does not sequester endogenous Prox1? Might cellular expression of anti-Prox1 and binding to Prox1 stimulate Prox1 degradation or block the epitope used to detect endogenous Prox1? Stating anti-Prox1 only sequesters extracellular Prox1 may not be correct and data showing that Prox1 has only extracellular activity, but not intracellular activity is needed. It is also important to show/identify the cells expressing AAV delivered anti-Prox1 antibody so they can be related to the changes in Prox1 expression that is noted. What is the promoter driving aProx1 in the AAV vector? Other measures of stopping protein exchange/extracellular transport could be used to confirm Prox1 secretion from BCs is necessary for its effect. These concerns come with the unchanging high PROX1 expression in other neurons; it seems unlikely that these expressions aren't being affected by the authors' attempts at disrupting PROX1 or they are not affecting MG PROX1 in any way.

6. Page 8 first full paragraph: With MNU, the MG cluster increases many fold in size (ED10). However, based on EdU immunostaining and general MG immunostaining such an increase in MG number appears impossible. In fact, there appears to be a significant decrease in the rPR cluster after MNU that also cannot be accounted for. How is this tension resolved?

7. Page 10, first paragraph: as mentioned in point #4, relative measures are sometimes inadequate. Using ONL to INL ratios to denote degeneration is problematic due to possible differences in INL thickness during degeneration. In these cases, absolute measures of layer thickness are required.

8. Page 10, second paragraph: The authors inject the AAV2-aProx1 disruptor at P10 in the rd10 mice. While the authors are rightfully concerned with expressing the PROX1 antibody at a timepoint where degeneration has not progressed too far, this leaves the results incomparable to the non-degenerative as well as the Rp1 model since MG reprogramming potential changes with age. Later time points in rd10 and earlier time points in Rp1 should be used to gauge if there could be any overlap in age.

9. Page 9, first full paragraph: ED12 shows that MGPC, cluster 20, specific genes, while higher in cluster 20, are also extensively expressed in other clusters; why is this occurring even though cluster 3 should be resting MG. How does this categorization of MG and MGPCs compare overall to the Blackshaw data?

Minor Concerns

10. For all the immunofluorescence images, it would be helpful to the reader to label the panels for which proteins/genes are being assayed. It would also be helpful if experimental timelines were included in panels where appropriate.

11. Fig. 1 and ED S1 indicate that retinal injuries from MNU, NMDA, light, and DTA expression all induce Prox1 expression in MG; however, only MNU stimulates delamination of the MG. This should be addressed in the manuscript.

12. The introduction appears to be shorter than expected. While the authors go through background information and previous findings, both their hypothesis and a synopsis of their findings is missing.

13. ED. S1d: colors in key do not completely match colors in graph.

14. Figs. 1-3, and extended data Fig. 3: it is not clear what the "4 dpi" under "vehicle" heading means. If this is an uninjured control, this labeling is misleading. Should it only be in the MNU-treated retina panels?

15. Page 5, first paragraph: While figure ED2 was cited as NMDA/light damage results, ED2a and ED2b appear to be the Blackshaw scRNAseq data for mouse MG development. Thus, no UMAP presentation of Prox1 is shown after injury.

16. Page 5, first paragraph: ED2d shows zebrafish ascl1a expression somewhat inconsistent with its known behavior. Control MG appear to show high expression while certain injury time points show lower expression than expected. The way these figures were obtained (ED2c and ED2d) should be explained.

17. Page 6, first paragraph, last sentence: "Therefore, these results suggest that the decrease in Prox1 synthesis in BCs, rather than the death or functional impairment of the cells, leads to a reduction in the amount of Prox1 transferred to MG." The data simply indicates normal cell numbers and normal visual optomotor response, but whether BCs are impaired in some way when Prox1 is deleted remains a possibility.

18. Page 6, bottom paragraph: FLAG-peptide is used as control for PROX1 protein as well as AAV injections, however, flag staining should also reveal flag peptide such as in figures 2f and ED6, but this is not the case. A reason should be given.

19. Page 6, bottom paragraph: EdU measures such as in figure 2h show EdU positive MG of non-canonical shape (such as

ones with horizontal nuclei seemingly outside of the INL). While this is not necessary a critique of a paper, discussions as to why this is would be interesting. Could these have been mislabeled?

20. Page 7, top paragraph: The PROX1 peptide injection into the zebrafish retina seems to show some images of decreasing MG amount (figure ED7d). A measure either in cell number or cell death could eliminate the confound of increased apoptosis. In ED7L, what are the numerous cells labelled as “others?”

21. Extended data Fig. S7: gfap:GFP fish exhibit GFP throughout the MG processes and therefore it is very difficult to conclude flag-Prox1 is actually in these cells. A nuclear marker for MG, like Sox2 (used in Fig. 1) would be better for showing co-staining and quantification. Panel d: the mouse monoclonal anti-Ascl1 antibody is not typically used to detect Ascl1a in the zebrafish retina, please provide validation to its specificity for detecting Ascl1a expression. Panels k and l: The EdU-based lineage tracing data is unclear and confusing. When after injury was EdU given and when were retinas harvested for analysis. A timeline of the experiment would help. Finally, based on this figure (panels k and l) it is stated in the text (page 7 top paragraph) that: “Consequently, the regeneration of retinal neurons, identified by their incorporation of EdU and loss of MG-specific gfap-EGFP expression but gain of cell type-specific marker expression, was diminished in the injured zebrafish retinas injected with FLAG-Prox1”. However, the data seems to indicate that there is no significant effect of Flag-Prox1 on regeneration of most retinal neuron types in the retina as indicated in panels k and l. Please clarify.

22. Page 8 bottom paragraph: Despite the Hes1 expressing cluster 20 in the scRNAseq data being a relatively small fraction of the whole, HES1 staining in figure 3k show extensive high expression of HES1 throughout the retina. How can this discrepancy be explained?

23. Page 9, first full paragraph: Many of the genes being shown in figure ED12 have deceptively low percentage expression. Some percentage/absolute cell expression as in figure 3 could be helpful.

24. Page 9, bottom paragraph: figure 3k,l,m experimental timelines are missing.

25. Page 9, bottom paragraph: The authors state that PROX1 blocks the conversion of MG to proliferative MGPCs; however, the authors have not characterized the resulting proliferative cells enough to state that MGPCs have been established based on PROX1 disruption. Also, the authors state that PROX1 suppresses the expression of target genes such as Notch1 and Ccnd1. While the authors have shown that PROX1 disruption leads to increased Notch1 and Ccnd1 expression. The authors have not established a direct, targeted effect between Prox1 and the downstream genes. Word choice is important here.

26. Page 11, top paragraph: Figure 4h is confusing. Even though the y axis label states EdU+Tdtomato+ cell counts, Tdtomato- measures are included in the graph. These graphs should be separated, or the labels should be corrected.

27. Page 11, top paragraph: This is an extension of point #1. Around 1 EdU+ cell co-labeling with rhodopsin per $2 \times 10^4 \mu\text{m}$ lead to a significant increase in visual acuity. Is this expected?

28. Page 11, bottom paragraph: Figure ED15a, c denotes the timeline of the extended EdU rd10 experiment, but why are the AAV injection times, tamoxifen injection times, and EdU injection times so much different than anything else, especially to the comparable rd10 normal experiment of figure 4f? In addition, the analysis timing is given as a range of days. Are the results a mixture of these date ranges or a specific date in these ranges? If mixtures are used, why are they grouped together rather than being used as a time course?

29. Page 11, bottom paragraph: in figure 5a and c, the graphs show a steady decline in PROX1 measures while the PROX1 levels across different time points in the pictures appear to be similar. Maybe other pictures could better represent these differences across time points.

30. Page 12 top paragraph: Returning to point #2, the timeline given in figure 5c appears different from everything else. Why is so much time given between AAV injection and sacrifice without any EdU tracing even though the desired outcome is measuring the overall proliferation and lineage? Why are so few EdU injections given, and all right at the end? Assays of neuronal differentiation 5-9 days post EdU injection may not be sufficient time for proliferating cells to differentiate and it would not reveal if these cells survived in the retina for prolonged periods. Additional time points post EdU should be explored as should survival of regenerated neurons. Also, are regenerated neurons functional?

31. Page 12 top paragraph: The methods used to obtain visual acuity measurements as in figure 5f, while given in materials and methods, should be mentioned in the main results body.

32. Page 15 bottom paragraph: The “area” measurement given in figure ED18c, while relative, should have some explanation behind it. Is this a measurement of pixel density, intensity, or a combination thereof?

33. Three different Prox1 antibodies and 2 different Sox2 antibodies are indicated in Table 1. Please indicate which antibodies were used in each experiment and why they were chosen.

34. In the Discussion it is proposed that Prox1 depletion in Muller glial cells stimulates their transition to a progenitor via Notch signaling and these progenitors then proliferate; however in the zebrafish retina Notch signaling inhibits injury-dependent Muller glia proliferation and in the mouse retina it was reported in a biorxiv preprint from Hoang lab (Robust reprogramming of glia into neurons by inhibition of Notch signaling and NFI factors in adult mammalian retina) that Notch also inhibits Muller glia cell reprogramming and proliferation in the mouse retina. This information should be included in the Discussion.

Reviewer #2

(Remarks to the Author)

In this manuscript Lee, Kim and colleagues investigate the role of Prox1 in limiting the neuro-regenerative potential of Müller glia (MG) following retinal injury. They demonstrate that Prox1 accumulates in MG following retinal injury in mice, but not in regenerative-competent zebrafish. The source of Prox1 is exogenous to MG. Depleting Prox1 or blocking its uptake restores proliferative potential to MG following injury. This has limited effectiveness in severe, early onset degenerative disease, but is able to prevent degeneration and visual decline in late onset disease when delivered prophylactically. This study represents a considerable amount of work with hypotheses tested in multiple models and mechanistic experiments giving a strong, robust manuscript. The potential for neuroprotection/recovery in retinitis pigmentosa and the further dissection of

mechanisms governing MG reprogramming is of importance to the field.

The following considerations are needed to improve the manuscript:

1. I agree with the suitability of the heading "Delaying vision loss in early onset retinitis pigmentosa mouse models by blocking Prox1" but find that some of the description of the results does not match this heading and somewhat oversells the magnitude of effects e.g. "Significant numbers of EdU-labeled newborn Rhodopsin-positive rPRs were found in the ONL of AAV2- α Prox1-infected Pde6brd10/rd10;Glast-CreERT;R26tdTom/+ mouse retinas (Fig. 4f,g)". This does not appear to be supported by any quantification. Indeed, numbers of EdU(+);tdTom(+) cells appear very low (mean $< 2 / 2 \times 10^4 \mu\text{m}^2$). Similarly, the ONL thickness increase and scotopic a-wave recovery are extremely modest, rather than "remarkably elevated". The language describing the results in this section should be more carefully considered. It is unsurprising that the effect is very limited given the harsh degenerative environment of the model and the persistence of the rd10 mutation. This reality is encapsulated nicely in introductory sentence to the following section.
2. The number of mice used for visual acuity in Figure 5G is very low ($n = 2$). Treated mice only go to 8 months of age. While at this time-point there is complete visual recovery, it is unknown if this persists or would diminish over time as in the untreated mice which show further decline from 11 months. It should be noted that with the current data, a conclusion on the long-term benefits of this treatments cannot be made.
3. Extended data 7K. Number of EDU+ Sox2+ cells is significantly reduced with FLAG-Prox1 but this is a limited reduction of roughly 30%. Data in 7I suggest that the proportion of EDU+ Sox2+ cells is increased, driven by an equally large loss of EDU in other cells (red). What is the identity of these cells, suggest that Prox1 limits proliferation of more than just MG in this context?
4. The author's hypothesis that chondroitin sulfate proteoglycans facilitate Prox1 transfer could be tested by intravitreal injection of Chondroitinase ABC after NMDA injury in mice and quantify the effects on Prox1 transfer to MG.
5. Introduction is missing a clear hypothesis or aim, not clear what knowledge gap is being addressed from this work for the general reader

Minor considerations

1. What is the significance of the increase of EdU-labeled microglia in the injured retinas of Prox1 $^{fg/fg};\text{Chx10-CreERT2}$ mice? Does MG proliferation result in reduced MG phagocytic capacity and a resultant increase in microglia to fill this role?
2. Pg 20. "resuspended in resuspended in the resuspension buffer"

Reviewer #3

(Remarks to the Author)

In this study, the authors demonstrate the novel importance of Prox1 in rendering retinal MG incapable of reentering the cell cycle and becoming MGPCs. Prox1 accumulates in MG in retinal degenerations and was seen to originate from neighboring neurons, in this case they primarily studied Prox1 from bipolar cells. Deletion of bipolar cell Prox1 gene or sequestration of extracellular Prox1 protein by an antibody induced the reprogramming of MG into proliferative retinal progenitor cells in injured retina. Using these findings the authors successfully delayed vision loss in mouse models of RP. It is a very interesting study that should be of significance, and well written. The finding of intercellular Prox1 transfer is novel and the delay in vision loss is striking, particularly in the late onset model. I only have a few points.

- (1) The ending of the Introduction feels very abrupt and seems like it needs a hypothesis or aim statement.
- (2) Supp Fig 1d has incorrect figure legend colors.
- (3) It would be nice to have seen electron microscopy images to see how much of the photoreceptor structure is preserved in the AAV2- α Prox1 infected RD10 mice.
- (4) In Fig 2e the authors have counted the number of Sox2+/Edu+ cells but what proportion of MG become Edu+. I think this proportion is important to show, to highlight the effectiveness of eliminating Prox1. What proportion of MG become reprogrammed and turn into other cell types?
On a similar note, what proportion of Edu+ cells become the other different cell types? (e.g., what proportion of Edu+ cells colocalize with Iba+ or PKCa).

Version 1:

Reviewer comments:

Reviewer #1

(Remarks to the Author)

Lee et al., did a good job in revising their Prox1 paper. However, a relatively small number of comments remain to be addressed.

Major adjustments to the paper

1. Transdifferentiation vs neuroprotection: Although the authors demonstrate little or no changes in neuroprotection upon Prox1 suppression, they do not demonstrate a lack of transdifferentiation as claimed. Selecting tdT+ cells that are sox2- and EdU- for this analysis may result in a bias in your analysis. It is clear from the research papers reported from the Reh and Blackshaw labs that EdU+ and Sox2+ MG can transdifferentiate. Just because a MG cell undergoes cell division does not mean it is a neuronal progenitor. Furthermore, Notch inhibition and NFi suppression are only a couple of ways to stimulate transdifferentiation. Other ways include Ascl1 overexpression and Trichostatin A treatment as shown by the Reh lab. Thus, there are likely many ways to stimulate transdifferentiation. I think the authors should rely more on their scRNAseq data sets to confirm or refute MG transition to a proliferating retinal progenitor state and to inform the reader as to the extent this

transition occurs by comparing to scRNAseq data previously obtained using RNA from the developing retina (Hoang, Blackshaw data).

2. Regarding Chx10 promoter: although you report in your rebuttal that Chx10 promoter is expressed in ~20% of MG, you also state in the text (line 182) it is BC specific. "...CreERT2 selectively in the BC population in a Tam-dependent manner ..."

The text should be edited for accuracy. In addition, because Prox1 RNA is expressed in MG, the expression of CreER in MG, even at low levels, is likely to reduce this expression thus confounding the conclusion that MG Prox1 is derived from BP cells.

3. Reviewer #1's figure 4 as well as the explanation of the extracellular nature of Prox1 a/Prox1 should be included in the main body. Same for Reviewer #1 figure 5.

4. Regarding Reviewer#1 comment 18 and response highlighted in text on page 8. ED6 indicates the FLAG-peptide remains in the vitreous of the injured retina while the FLAG-Prox1 can move from the vitreous into the injured retina. What is the explanation for this different behavior of the 2 proteins? If the FLAG-peptide never enters the retina, it may not be a good control for the FLAG-Prox1 that does enter the retina.

Minor adjustments

1. Figure 1d is not cited in the paper

2. Ext Fig 9c is not cited

3. No quantification of EdU+ or Ascl1+ cells at 4dpi in the zebrafish after Prox1 protein injection and MNU injury.

4. Fig 6a: Are uninjured bipolar cells less in number compared to the injured, or is this a case of relative abundance?

5. Fig7h: The existence of EdU+Tdt- cells should be explained with the Glaxt-Cre quantification provided in the review where only ~60% of Sox2+ MG were labeled with Tdtomato.

6. Ext fig 8a: Why do separate staining of the Prox1 a-Prox1 pulldown using Prox1 vs HA antibody produce drastically different sized bands?

7. Ext fig 8: In the legends "they receive the HP from HP" requires clarification of both the sentence and HP.

8. Page 3 line 75 to 77: "Notch signaling is also activated... neuronal differentiation." The line appears confusing in stating that Notch is activated but should go down. Clarification is required in stating differential Notch factor expression is required for zebrafish MG regeneration.

9. Page 4 line 83 to 85: "further support this process... TAZ" Suggest replacing "which suppresses yes-associated protein" with "leading to the disinhibition of yes-associated protein" to avoid confusion in this phrase either modifying "the inactivation" or "Hippo signaling."

10. Page 16 line 429 Extra: "in" or replace with "in the mouse retina" in the middle of the sentence "the number of EdU-labeled cells in remains lower..."

11. Page 21 line 580 to 587: The production and injection of control FLAG peptide was not described in the methods.

12. Page 22 line 589 to 597: AAV plasmids, production, and controls used were not described in the methods

13. Page 5 line 130: missing "is" in sentence "where selective PR degeneration induced by DTA."

14. Page 9 line 247: "We also observed a distinctive lineage from to the BC..." Requires some editing.

Reviewer #2

(Remarks to the Author)

The authors have addressed all of my initial comments

Reviewer #3

(Remarks to the Author)

The authors have adequately answered all the previous comments.

Version 2:

Reviewer comments:

Reviewer #1

(Remarks to the Author)

The revised manuscript has addressed all my concerns.

Reviewer #1 (Remarks to the Author):

Summary

The authors set out to restore lost photoreceptors and visual acuity after retinal injury and degeneration by inducing mouse Müller glia (MG) proliferation and neurogenesis via the disruption of PROX1 accumulation.....

Major Concerns

Overall: **The impact of the PROX1 induced regeneration, the crux of the paper, appears to be relatively small.** In cases of non-relative measures, less than 5 EdU cells per $10^4\mu\text{m}$ were observed after PROX1 disruption. Yet this led to significant ONL preservation as well as vision restoration. This could be due to **the arbitrary amount and timing of the EdU injections failing to capture the real amount of proliferating MG** (although this could be partially resolved by the quantification of coexpression of EdU-Tdt+ cells with neuronal markers). **More measured EdU injections should be done to capture more proliferating cells.**

Re: We appreciate the reviewer's comment regarding the potential underestimation of newborn cells due to the EdU labeling protocol.

In our study, EdU was administered daily for 4 consecutive days following MNU injury (Fig. 3b, 4b). Extending this period was constrained by the limitations of the MNU injury model, which we analyze the retinas at 4 days after the injury.

For RP model mice, we labeled cells born within the last 7 days prior to sample collection. In the *Ped6b^{rd10/rd10}* model, where newborn photoreceptors degenerate within 7 days (Extended Data Fig. 16), extending EdU labeling beyond this period would likely yield limited additional insights.

For *Rp1^{m/m}* mice, which exhibit a slower degeneration than *Ped6b^{rd10/rd10}* mice, we extended EdU injections to 4 weeks. This revised protocol resulted in an increase in the number of EdU-labeled cells, as presented in the updated Fig. 8, d-f. However, even with these adjustments, the number of EdU-labeled cells remains lower compared to zebrafish following MNU injury (Extended Data Fig. 7k).

These findings suggest that blocking Prox1 transfer alone may be insufficient to fully restore regenerative capacity to that observed in zebrafish. Additional events, such as Notch inhibition or Yap/Taz activation following post-reprogramming, may be needed to promote robust MG or MGPC proliferation. This consideration has been incorporated into the Discussion (highlighted in page 16), and will be a focus of our future studies.

The prospect of transdifferentiation and neuroprotection are also not really explored in this study.

Re: We appreciate the reviewer's suggestion to explore the potential transdifferentiation of MG into neurons and the neuroprotection in the mouse retina with Prox1 transfer blockage.

1) To assess MG-to-neuron transdifferentiation, we analyzed tdTom(+);Sox2(-);EdU(-) retinal cells, which may represent neurons transdifferentiated from MG or regenerated from MG prior to the EdU-labeling period. In AAV-anti-Prox1-infected *Ped6b^{rd10/rd10}* mouse retinas, only 1 among 355 tdTom(+) cells met this criterion (highlighted in page 12). These findings suggest that MG-to-neuron

transdifferentiation unlikely occurs under our experimental conditions.

Effective transdifferentiation of MG into neurons typically requires both Notch signaling inhibition and loss of Nfi transcription factors, which maintain MG identity (Clark et al., 2018; Le et al., 2024). In Prox1 transfer-blocked mouse retinas, Notch1 and Hes1 were expressed in the RPC population (Fig. 5e–j, 6f–h), and no significant changes in Nfi expression were observed in MG (data not shown). Thus, Prox1 depletion appears to reprogram MG to RPCs rather than facilitating transdifferentiation into neurons.

2) To investigate neuroprotection, we analyzed TUNEL(+) apoptotic cells in AAV-anti-Prox1-infected *Ped6b^{rd10/rd10}* and *Rp1^{m/m}* mouse retinas and compared them with controls. As shown in Extended Data Fig. 14e (rd10) and 17e (Rp1), there was no significant reduction in TUNEL(+) cells, suggesting that the observed increase in outer nuclear layer (ONL) cells in these models is unlikely due to neuroprotection. Similar results were observed in *Prox1^{fg/fg};Chx10-CreERT2* mice. Due to space limitations, we, however, did not include these findings in the manuscript but have provided them only for the reviewer's inspection (Reviewer #1's inspection fig. 1).

[figure redacted]

Reviewer #1's inspection fig. 1. Comparison of cell death in MNU-injured *Prox1^{fg/fg}* and *Prox1^{fg/fg};Chx10-CreERT2* littermate mouse retinas. (a) *Prox1* was selectively deleted to express EGFP complementarily in BCs of *Prox1^{fg/fg};Chx10-CreERT2* mouse retinas through repeated injections of Tam, which activates CreERT2 recombinase expressed in the BC population. The mice were then injected with PBS or MNU into the peritoneal space of the mice. 20 ng FLAG-Prox1 recombinant proteins were injected into the mouse eyes as indicated. Distribution of TUNEL-positive apoptotic cells and EGFP-positive *Prox1-cko* BCs in the retinas of *Prox1^{fg/fg}* and *Prox1^{fg/fg};Chx10-CreERT2* littermates was visualized. (b) Numbers of TUNEL-positive apoptotic cells in the indicated mouse retinas are presented in the graph (data from 3 independent litters).

2. Figure 1 and figure ED1 are supposed to compare different injury paradigms and PROX1 accumulation. **Please include experimental timeline for these figures and all subsequent figures where appropriate.** Fig. 1a and 1b show days post MNU injury while 1d-g neglect to mention which of these MNU time points were used.

Re: We appreciate the reviewer's suggestion to improve the clarity of our figures by including experimental timelines. In response, we have added specific post-injury timelines in all relevant panels in Figures, as well as Fig. 1 and Extended Data Fig. 1, to specify which MNU time points were used.

Also, **comparison of cell death between the different injury paradigms is important in order to correlate their cell death with MG activation/proliferation.**

Re: In response to the reviewer's suggestion, we have added TUNEL assay results to show apoptotic cell detection for all relevant figures, including Fig. 1 and Extended Data Fig. 1.

3. A couple of inducible Cre mouse models such as *Glast-CreER* and *Chx10-CreERT2* were used in the paper. Although these mouse lines were characterized in previous papers, much of the authors' results rely on their expression specificity after crossing them with new lines; thus, **expression specificity and recombination efficiency must be**

established in the paper.

Re: We agree with the reviewer that demonstrating the expression specificity and recombination efficiency of the Cre mouse models used in our study is needed. To assess the activity of Glast-CreER in MG, we employed the tdTom Cre reporter in *Glast-CreER;R26^{tdTom}* mouse retinas. Our findings show that approximately 56% of Sox2-positive MG were tdTom-positive, indicating CreER activity for gene manipulation in majority MG (Reviewer #1's inspection fig. 2b). Moreover, all tdTom-positive cells were Sox2-positive MG, and none of the Vsx2+;Sox2- BCs were tdTom-positive, demonstrating that Cre recombination occurred selectively in MG (Reviewer #1's inspection fig. 2c). This analysis confirms the specificity and efficiency of Cre recombination in MG, supporting the reliability of our experimental model. Due to space constraints, we have not included this data in the manuscript but have provided it for the reviewer's inspection (Reviewer #1's inspection fig. 2, a-c).

[figure redacted]

Reviewer #1's inspection fig. 2. Efficacy and specificity of Cre recombinase in *Glast-CreER* and *Chx10-CreERT2* mice. (a,d) P30 *Glast-CreER;R26^{tdTom/+}* mice (a) and *Chx10-CreERT2;R26^{tdTom/+}* mice (d) were injected with Tam at the indicated time schedule and the eyes were isolated from the mice for the immunohistochemical analysis at P40. Retinal cells affected by Cre recombinase were monitored by visualizing tdTom Cre reporter. MG and BC identities of the tdTom-positive cells were determined by co-staining of Sox2 and Vsx2, respectively. Given the co-expression of Sox2 and Vsx2 in MG, Vsx2+;Sox2- cells were designated to BC. (b,e) Population of tdTom-expressing cells among Sox2+ MG (b) and Vsx2+;Sox2- BC (e) were determined and shown in the graphs. (c,f) Vsx-;Sox2+ or Vsx2+;Sox2+ MG and Vsx2+;Sox2- BC population among tdTom-positive cells were determined and presented in the graphs.

This leads to problems such as questioning the bipolar cell specific nature of the Chx10 mouse. According to the Blackshaw data used in the paper, Chx10 is not bipolar specific in the mouse retina. In fact, the **mouse MG appear to express Chx10**. The exogenous nature of PROX1 then becomes muddled.

Re: As noted, Vsx2 (formerly Chx10) can indeed be detected in MG, but its expression level is significantly lower than that in BCs (Reviewer #1's inspection fig. 2a,d). In our *Chx10-CreERT2;R26^{tdTom}* mouse retinas, we found that approximately 80% of tdTom-positive cells were Vsx2+;Sox2- BCs, with only about 20% being Vsx2+;Sox2+ MG, demonstrating that Cre recombination predominantly occurred in BCs (Reviewer #1's inspection fig. 2f). We also provide the results for the reviewer's inspection (Reviewer #1's inspection fig. 2, d-f).

4. The paper relies on relative measures throughout. **PROX1 accumulation is measured through fluorescent amount/intensity relative to neurons**, even though PROX1 expression in neurons across different models and paradigms are not guaranteed to be comparable.

Re: We appreciate the reviewer's concern about using relative measures to assess Prox1 accumulation in MG. As with other markers, it is difficult to achieve identical immunostaining intensities across samples, especially under varied experimental conditions. To address variability, we normalized the Prox1 signal using an internal control unaffected by experimental conditions. Specifically, we used Prox1 immunostaining in BCs, where expression remains stable after various retinal injuries, as a reference (Extended Data Fig. 1d).

We, however, realized that this normalization approach may appear more

complex than a straightforward intensity comparison between samples, potentially leading to confusion for some readers. To address this, we have revised the figures to include graphs displaying y-axis values of "Prox1 intensity relative to control samples," with normalization based on Prox1 intensity in BCs for each sample.

This approach is analogous to using beta-actin in western blot analyses to normalize protein band intensities, correcting for sample variability. By normalizing Prox1 intensity in MG to the stable Prox1 signal in BCs, we ensured that any potential differences in staining intensity due to technical variations (e.g., over- or under-staining) are accounted for. This method provides a reliable and reproducible measure of relative Prox1 expression, reducing the risk of misinterpretation across different models and injury paradigms.

Dkk3-Cre is used as a control comparison in figure 1h and ED3b even though such a mouse line is never mentioned in the body or the reason for the comparison given.

Re: Due to spatial limitations of the initially submitted manuscript, we were unable to provide a detailed rationale for the use of the *Dkk3-Cre* mouse line in the main text. Instead, we had included an explanation of the *Dkk3-Cre* line and its role in relative measurements in the Methods section. To improve clarity, we have now added the explanation for the mouse line in the main text as "As anticipated, robust EGFP expression was observed in HCs, BCs, and AC subsets in *Prox1^{fg};Dkk3-Cre* mice, where the *Prox1* knock-in allele was eliminated in embryonic RPCs and their descendant retinal cells ⁴¹ (Extended Data Fig. 3)" (page 7, highlighted). This addition should ensure readers understand the rationale behind using this mouse model for control comparisons.

Retinal degeneration is measured through ONL to INL thickness comparison in figure 4, 5, and on. Even though relative measures are important and sometimes the most telling measures, **absolute measures are also sometimes required to give a better understanding of the magnitude of the changes observed.** This leads to the next major concern.

Re: We appreciate the reviewer's concern regarding the use of relative measures for retinal degeneration. Absolute measures of retinal thickness can indeed provide important insights into the progression of retinal degeneration, but in the context of retinal sections, they can be highly variable due to the position of the section within the eye. For example, central retinal sections are generally thinner than those from the periphery as illustrated in Reviewer #1's inspection fig. 3. This positional variability can lead to inconsistent absolute thickness measurements across different sections.

To minimize this variability and the risk of misinterpretation, we opted to use relative measures by comparing the ratio of ONL to INL thickness within the same section. This relative comparison remains consistent regardless of the section's position within the healthy eye, allowing for a more reliable assessment of retinal degeneration across different samples. This approach reduces the confounding effects of anatomical differences between central and peripheral retinal sections and ensures more meaningful comparisons between eyes from different experimental groups.

[figure redacted]

Reviewer #1's inspection fig. 3. Diagram depicts retinal thickness differences depending on the section position.

Retinal thickness of cross-section of central eye cup is thinner than those of intermediate or peripheral eye cup sections. However, the ratio between ONL and INL in each section is not changed.

5. Page 7 bottom paragraph: The sequestration of extracellular PROX1 using anti-PROX1 antibody in figure 2 leads to the question of whether such measures (including Chx10 based recombination) actually eliminated the transfer of PROX1 from exogenous sources into MG in vivo. It is stated that anti-Prox1 is acting in the extracellular space to inhibit Prox1 uptake by Muller glia. Please **provide data indicating cellular expression of anti-Prox1 does not sequester endogenous Prox1.** Might cellular expression of anti-Prox1 and binding to Prox1 stimulate Prox1 degradation or block the epitope used to detect endogenous Prox1?

Re: We appreciate the reviewer's concern regarding the potential effects of anti-Prox1 on endogenous Prox1 expression. To address this, we prepared AAV construct and took several measures to ensure that anti-Prox1 expressed in retinal neurons does not interfere with endogenous Prox1 expression:

1) Secretion and Expression of Anti-Prox1: We utilized an IL2 signal sequence in the anti-Prox1 antibody cDNA to promote secretion through the ER and Golgi while minimizing cytoplasmic expression of the antibody within the neurons. This strategy was designed to reduce potential interference of endogenous Prox1 with the anti-Prox1 antibody in the cytoplasm.

2) Evaluation of Prox1 Expression: To assess whether anti-Prox1 affects endogenous Prox1 levels, we compared Prox1 immunostaining intensity between cells expressing anti-Prox1 and those do not (see revised Fig. 4c and Extended Data Fig. 9a,b). The results showed that Prox1 intensity in retinal neurons, such as BCs and ACs, expressing anti-Prox1 was not significantly different from that in neighboring cells lacking anti-Prox1. This suggests that anti-Prox1 expression does not substantially alter endogenous Prox1 levels in the retinal neurons.

Stating anti-Prox1 only sequesters extracellular Prox1 may not be correct and **data showing that Prox1 has only extracellular activity, but not intracellular activity is needed.**

Re: We found that injection of rabbit anti-Prox1 IgG into the vitreous body of mouse eyes led to a decrease in Prox1 intensity in MG (see Reviewer #1's inspection fig. 4). Since IgG cannot penetrate cells across plasma membrane unless endocytosis of IgG-bound membrane receptors occurs, this effect is likely due to anti-Prox1 binding extracellular Prox1.

Additionally, any anti-Prox1 antibody inadvertently released into the cytoplasm of infected cells would be unlikely to bind intracellular Prox1. This is because the reduced conditions of the cytoplasm prevent the formation of intramolecular disulfide bridges, which is required for the antibody's stability and binding capacity (Wirtz and Steipe, 1999, *Protein Sci*, 8: 2245–2250).

These findings suggest that the decrease in Prox1 in MG is mediated through the sequestration of extracellular Prox1 by anti-Prox1 secreted from retinal cells infected with AAV2-anti-Prox1 rather than the anti-Prox1 antibody affecting the intracellular Prox1 in the infected cells.

[figure redacted]

Reviewer #1's inspection fig. 4. Decrease of Prox1 in MG of MNU-injured mouse retinas by intravitreal injection of anti-Prox1 antibody. (a) Tam was injected into the peritoneal space of *Glast-CreER;R26^{tdTom/+}* mice at P14 to activate CreER in MG in the mouse retinas. MNU was injected into the peritoneal space of the mice at P30 to degenerate PRs in the retinas. Concomitantly, rabbit immunoglobulin G₁ (Rb IgG) or rabbit anti-Prox1 polyclonal antibody (Rb α Prox1) was injected to the intravitreal space of the MNU-injected mice. The eye sections were prepared from the mice at 4 days after the MNU injury. Distribution of Prox1 in the mouse retinal sections was visualized by immunostaining. The activation of CreERT in MG was determined by detecting Cre reporter tdTom. Nuclei of retinal cells in the sections were visualized by DAPI staining. tdTom-positive MG cell bodies are outlined by dotted lines. (b) Prox1 intensities of the tdTom(+) MG cells relative to those in Vehicle-injected control retina are plotted in the graph (data from 3 independent litters). *, $p < 0.05$; ****, $p < 0.001$; n.s., not significant (ANOVA test).

It is also important to **show/identify the cells expressing AAV delivered anti-Prox1 antibody** so they can be related to the changes in Prox1 expression that is noted.

Re: We have identified the majority of cells expressing anti-Prox1 are retinal ganglion cells (RGCs) and amacrine cells (ACs), which locate in GCL and lower INL, respectively (Extended Data Fig. 9c).

What is the promoter driving aProx1 in the AAV vector?

Re: We apologize for the oversight. The AAV2 vector used for delivering anti-Prox1 was driven by the EF1A promoter. This information has been added to the revised Supplementary Table 1.

Other measures of stopping protein exchange/extracellular transport could be used to confirm Prox1 secretion from BCs is necessary for its effect.

Re: At present, our available methods for studying Prox1 transport involve either deleting the *Prox1* gene in the donors (as in *Prox1^{fl/fl};Chx10-CreERT2* mice) or sequestering Prox1 in the extracellular space using anti-Prox1. We do not have alternative methods to specifically block Prox1 transfer from BCs to MG beyond these approaches.

These concerns come with **the unchanging high PROX1 expression in other neurons;** it seems unlikely that **these expressions aren't being affected by the authors' attempts at disrupting PROX1** or they are not affecting MG PROX1 in any way.

Re: As previously explained, our approach with AAV2-anti-Prox1 aimed specifically to reduce extracellular Prox1, which becomes the exogenous Prox1 in MG through surface binding and penetration, without affecting endogenous Prox1 levels in retinal neurons. We achieved this by showing that Prox1 levels in retinal neurons remained unchanged (Extended Data Fig. 9a,b), while Prox1 levels in MG were

significantly reduced following AAV2-anti-Prox1 infection (Fig. 4c,d). This evidence supports that our methods effectively targeted extracellular Prox1 while leaving neuronal Prox1 levels unaffected.

6. Page 8 first full paragraph: **With MNU, the MG cluster increases many fold in size (ED10)**. However, based on EdU immunostaining and general MG immunostaining such an increase in MG number appears impossible. In fact, there appears to be a significant decrease in the rPR cluster after MNU that also cannot be accounted for. **How is this tension resolved?**

Re: We appreciate the reviewer's observation regarding the apparent increase in the MG cluster size following MNU treatment, which seems inconsistent with general MG immunostaining results. MNU primarily targets and destroys rPR, which make up approximately 60% of the retinal cell population in uninjured mouse retinas. As a result, the relative proportion of other retinal cell types, including MG, increases significantly. This relative increase can be misinterpreted as an absolute increase in MG numbers when viewed in isolation from the loss of rPRs. In summary, the increase in the MG cluster size is likely due to the relative increase in MG proportion following the loss of rPR.

7. Page 10, first paragraph: **as mentioned in point #4, relative measures are sometimes inadequate**. Using ONL to INL ratios to denote degeneration is problematic due to possible differences in INL thickness during degeneration. In these cases, absolute measures of layer thickness are required.

Re: Please refer our response to the reviewer's comment #4.

Page 10, second paragraph: **The authors inject the AAV2-aProx1 disruptor at P10 in the rd10 mice**. While the authors are rightfully concerned with expressing the PROX1 antibody at a timepoint where degeneration has not progressed too far, this leaves the results incomparable to the non-degenerative as well as the Rp1 model since MG reprogramming potential changes with age. **Later time points in rd10 and earlier time points in Rp1 should be used** to gauge if there could be any overlap in age.

Re: We selected postnatal day 10 (P10) for AAV2-anti-Prox1 injection in rd10 mice to specifically target post-mitotic cells, minimizing dilution of the plasmid due to cell division since retinal neurogenesis is largely complete by this age. While we recognize that this timing limits direct age-matched comparisons with non-degenerative and other models such as Rp1, it represents the earliest feasible time point for stable AAV2-anti-Prox1 expression, given the two-week lag in protein expression following AAV infection.

To address the reviewer's suggestion, we also administered AAV2-anti-Prox1 to rd10 mice at 2 months of age and assessed visual acuity starting from 2.5 months. Contrary to the positive results observed with P10 injections, no visual recovery was noted in these older animals (see Reviewer #1's inspection fig. 5). These findings suggest that AAV2-anti-Prox1-induced retinal regeneration requires early intervention, prior to the completion of retinal degeneration, which aligns with an age-sensitive therapeutic window essential for efficacy.

[figure redacted]	Reviewer #1's inspection fig. 5. AAV2-αProx1 injection was not affected to late-stage RP model mice. Pde6b^{rd10/rd10} mice received intravitreal injections (Inj.) of AAV2 or AAV2-αProx1 (5X10⁹ vg/eye) at P60, when they have already lost their vision completely. Visual acuities of the injected mice were measured by Optomotry.
-------------------	---

9. Page 9, first full paragraph: ED12 shows that **MGPC, cluster 20, specific genes, while higher in cluster 20, are also extensively expressed in other clusters**; why is this occurring even though cluster 3 should be resting MG.

Re: Our clustering approach groups cells based on overall expression profiles and shared transcriptional features, rather than exclusively by specific marker genes. As a result, some RPC-specific genes may be expressed at varying levels in other clusters, even though these clusters are primarily classified based on distinct characteristics.

To assess the distinctiveness of cluster #20 (RPC) in comparison to other clusters, we calculated Pearson Correlation Coefficients (PCC) based on average expression values for clusters #3 (resting MG), #7 (activated MG), and #20 (RPC). The heatmap analysis demonstrates that the PCC between cluster #20 and cluster #3 was relatively low (0.65), and even lower between cluster #20 and cluster #7 (0.51). These findings indicate that cluster #20 is transcriptionally distinct from clusters #3 and #7 (Reviewer #1's inspection fig. 6a).

[figure redacted] Reviewer #1's inspection fig. 6. Re-evaluation of MG and RPC population. (a) Pearson correlation coefficients (PCC) on the average expression values among clusters #3 (resting MG), #7 (activated MG), and #20 (RPC) were calculated and presented in the heatmap. (b, c) MG clusters (cluster #3 and #7) and RPC cluster (cluster #20) extracted from our single-cell RNA-seq dataset were re-clustered by high variance genes in RPCs from the published data of Blackshaw group. Clusters were plotted by using UMAP (b) and PCA (c) dimension reduction methods, respectively.

How does this categorization of MG and MGPCs compare overall to the Blackshaw data?

Re: In the studies from the Blackshaw group, highly variable and strongly expressed genes in RPCs were identified (Clark et al., 2018), and zebrafish MGPCs were analyzed extensively (Hoang et al., 2020). However, the availability of MGPC data in mice is limited due to their inability to efficiently reprogram from MG to MGPCs under normal conditions.

To address this limitation, we utilized the high-variance genes identified in the Blackshaw dataset for RPCs to classify RPC populations in our single-cell RNA-

seq dataset. Specifically, we re-clustered the cluster #3, #7, and #20 from our data using these high-variance genes. The re-clustered UMAP visualization demonstrated clear separation of cluster #20 from clusters #3 and #7 (resting and activated MG). This separation was further corroborated by Principal Component Analysis (PCA) with 3D visualization (Reviewer #1's inspection fig. 6b,c), supporting our classification of cluster #20 as a transcriptionally distinct RPC population.

Minor Concerns

10. For all the immunofluorescence images, it would be helpful to the reader to **label the panels for which proteins/genes are being assayed**. It would also be helpful if **experimental timelines were included in panels where appropriate**.

Re: In the revised figures, we added the post-injury time labels. We also provide experimental timelines in the corresponding figures.

11. Fig. 1 and ED S1 indicate that retinal injuries from MNU, NMDA, light, and DTA expression all induce Prox1 expression in MG; however, **only MNU stimulates delamination of the MG**. This should be addressed in the manuscript.

Re: We observed that the extent of MG delamination is closely related to the severity of retinal injury. For example, high-dose MNU treatment leads to significant MG delamination, while lower doses of MNU result in only mild delamination (see Reviewer #1's inspection fig. 7). In similar, retinal injuries caused by light exposure and NMDA administration are less severe compared to MNU and DTA treatments (Extended Data Fig. 1). As a result, these milder injuries do not induce the same degree of MG delamination. We have included a statement in the manuscript to address this: "The accumulation of Prox1 in MG was more pronounced in MNU-treated and *Crx-CreERT2;R26^{DTA/+}* mouse retinas compared to light-injured retinas, where PR degeneration and MG delamination were less severe. These findings suggest that increased Prox1 expression in MG is a common response to retinal injury in mouse models, independent of the type of retinal cell degeneration but associated with the extent of injury" (page 6, highlighted).

[figure redacted]	Reviewer #1's inspection fig. 7. MNU dose-dependent delamination of MG. P30 mice were injected with MNU at the indicated concentration, and the expression of Prox1 in Sox2-positive MG cells were examined by co-immunostaining at 7 days post-injury.
-------------------	---

12. The introduction appears to be shorter than expected. While the authors go through background information and previous findings, both their hypothesis and a synopsis of their findings is missing.

Re: We apologize for the previous short Introduction section. In response to the reviewer's feedback, we have expanded the Introduction to include the following:

1) Background Information: We have provided a more detailed overview of the background and context of the study, elaborating on previous findings relevant to our research.

2) Hypothesis: We have explicitly stated our hypothesis, outlining the key assumptions and objectives guiding the study.

3) Synopsis of Findings: We have included a summary of our main findings and their implications, offering a clearer picture of the study's outcomes and relevance. These additions aim to give readers a comprehensive understanding of the research context, hypotheses, and conclusions. The revised Introduction now provides a more complete overview of the study.

13. ED. S1d: colors in key do not completely match colors in graph.

Re: We corrected the colors in the figure.

14. Figs. 1-3, and extended data Fig. 3: it is not clear what the "4 dpi" under "vehicle" heading means. If this is an uninjured control, this labeling is misleading. Should it only be in the MNU-treated retina panels?

The label "4 dpi" under the "Vehicle" heading was indeed misleading, as it suggested that this was related to an injury time point rather than a control condition. To clarify:

1) Correction of Labeling: We have corrected the labeling from "4 dpi" to "Day 4" to more accurately reflect that this time point corresponds to 4 days after the vehicle (buffer solution) injection. This change clarifies that the vehicle condition is a control and not an injury time point.

2) **Explanation in the Figures:** We have updated the figure legends and the corresponding panels to explain that the vehicle injection serves as a control to exclude non-specific effects of the buffer components, and that the “Day 4” designation refers to the time elapsed post-injection.

These modifications ensure that the labeling accurately represents the experimental conditions and avoids any confusion regarding the control and injury time points.

15. Page 5, first paragraph: While figure ED2 was cited as NMDA/light damage results, ED2a and ED2b appear to be the Blackshaw scRNAseq data for mouse MG development. Thus, **no UMAP presentation of Prox1 is shown after injury.**

Re: We apologize for the oversight in referencing the incorrect data. The figure legend for Extended Data Fig. 2 (ED2) indeed included information on both retinal development and injury models. To clarify:

1) **Updated Figures:** We have revised Extended Data Fig. 2 to include UMAP plots specifically showing Prox1 expression in MG after NMDA and light-induced retinal injury. These updated plots are now presented in Extended Data Figs. 2b and 2c.

2) **Additional Information:** The revised figures provide a clearer depiction of Prox1 expression and MG lineage following injury, addressing the concern about the absence of UMAP data related to Prox1 after retinal damage.

These updates ensure that the figure accurately reflects the data on Prox1 expression in the context of retinal injury.

16. Page 5, first paragraph: **ED2d shows zebrafish ascl1a expression somewhat inconsistent with its known behavior.** Control MG appear to show high expression while certain injury time points show lower expression than expected. **The way these figures were obtained (ED2c and ED2d) should be explained.**

Re: As we wrote in the corresponding figure legend, the data for Extended Data Figs. 2d and 2e were retrieved from a public database that includes both zebrafish development and injury models. We recognize that there appears to be inconsistency in the expression patterns of ascl1a, especially in NMDA-injury models. The differences in expression levels across control and injury time points may result from variations in dataset acquisition or annotation within the database. Further experimental validation, therefore, might be needed to resolve these inconsistencies.

17. Page 6, first paragraph, last sentence: “Therefore, these results suggest that the decrease in Prox1 synthesis in BCs, rather than the death or functional impairment of the cells, leads to a reduction in the amount of Prox1 transferred to MG.” **The data simply indicates normal cell numbers and normal visual optomotor response,** but whether BCs are impaired in some way when Prox1 is deleted remains a possibility.

Re: We acknowledge that our previous statement may have over-interpreted the data. We have revised the manuscript to remove the specific suggestion that Prox1 reduction in BCs is solely responsible for the decreased Prox1 transfer to MG (please see the last sentence of the first paragraph in page 8).

18. Page 6, bottom paragraph: FLAG-peptide is used as control for PROX1 protein as well as AAV injections, however, **flag staining should also reveal flag peptide such as in figures 2f and ED6, but this is not the case.** A reason should be given.

Re: FLAG-peptide was used as a control for FLAG-Prox1 in our experiments, but not

in AAV injections, where FLAG-tagged Ctrl-Ab was used as control. Unlike FLAG-Prox1, which can bind and incorporate into retinal cells, FLAG peptide alone does not penetrate across plasma membrane and is thus not retained within the retinal cells. Consequently, FLAG peptide is released into the vitreous humor and does not appear in retinal tissue. We have clarified this point in the revised manuscript, explaining that the absence of FLAG peptide staining in the retina is consistent with its expected release from the eye (highlighted in page 8).

19. Page 6, bottom paragraph: EdU measures such as in **figure 2h show EdU positive MG of non-canonical shape (such as ones with horizontal nuclei seemingly outside of the INL)**. While this is not necessarily a critique of a paper, discussions as to why this is would be interesting. Could these have been mislabeled?

Re: We apologize for the figure with not enough information. Given the massive migration of MG nuclei from the INL to the ONL in the injured mouse retina, we frequently observed Sox2-positive MG in the ONL of the injured mouse retinas, regardless of regeneration in the retina. However, EdU-positivity of the Sox2-positive cells in the ONL was only observed in AAV2- α Prox1-injected samples. Those EdU-labeled in the ONL of Ctrl Ab-expressing mouse retina were Iba1-positive microglia but not Sox2-positive MG or MGPCs. We, however, do not have a clear answer why those Sox2-positive newborn MG or MCPC nuclei remain in the ONL.

20. Page 7, top paragraph: **The PROX1 peptide injection into the zebrafish retina seems to show some images of decreasing MG amount (figure ED7d)**. A measure either in cell number or cell death could eliminate the confound of increased apoptosis. The image in the figure includes less EGFP-positive MG than the majority retinal parts.

Re: We have addressed the concern by replacing the previous image with a new one that includes a higher number of EGFP-positive MG. Additionally, we have conducted further analysis to measure MG cell numbers and assess cell death to rule out the possibility that the observed decrease in MG is due to increased apoptosis. These updated results and additional analyses are included in the revised manuscript.

In ED7L, **what are the numerous cells labelled as “others?”**

Re: The cells labeled as “others” in Extended Data Fig. 7I are negative for the specific markers listed in the figure. These cells could include retinal cells, including cone photoreceptors (cPR) and cone BC, and non-retinal origins, such as astrocytes and endothelial cells. We have added this clarification to the figure legend to provide more detail on the identity of these “others” cells.

21. Extended data Fig. S7: **gfap:GFP fish exhibit GFP throughout the MG processes and therefore it is very difficult to conclude flag-Prox1 is actually in these cells. A nuclear marker for MG, like Sox2 (used in Fig. 1) would be better for showing co-staining and quantification.**

Re: Given the penetration of Prox1 to the cytoplasm of MG and subsequent migration to their nuclei, we wanted to use the markers that are expressed in the entire MG area for the identification of exogenous FLAG-Prox1 in MG. Thus, we used GFP, which is expressed both in the cytoplasm and nucleus in MG of gfap:GFP

zebrafish retina (Extended Data Figs. 7, b and c). In addition, given the decrease of gfap:GFP signals along with transition of MG to RPC, we could also use GFP to identify Sox2+;GFP- MGPC from Sox2+;GFP+ MG. However, as the reviewer indicates, it would be more beneficial to use nuclear markers to quantify MG or MGPC numbers. Thus, we used Sox2 to quantify MG or MGPCs (Extended Data Fig. 7, f, k, and i).

Panel d: the mouse monoclonal anti-Ascl1 antibody is not typically used to detect Ascl1a in the zebrafish retina, **please provide validation to its specificity for detecting Ascl1a expression.**

Re: In previous papers, the critical roles of Ascl1 in MG reprogramming were investigated either by overexpression or knock-down approaches. The injury-induced elevation of Ascl1 expression in MG of zebrafish retina was also confirmed by analyzing *Ascl1* mRNA expression (including Fausett et al., J Neurosci (2008); Ramachandran et al., Nature Cell Biol (2010)) rather than examining Ascl1 protein expression. We, thus, had to tested with commercially available anti-Ascl1 antibodies to identify Ascl1 expression in gfap:GFP-positive MG lineage cells. Among five antibodies tested, only this antibody (Cat# sc-374104) works for the IHC.

REAGENT or RESOURCE	SOURCE	IDENTIFIER
Mouse monoclonal anti-ASCL1 (D-7)	Santa Cruz	sc-374104
Mouse monoclonal anti-MASH1	BD	556604
Mouse monoclonal anti-MASH1	invitrogen	14-5794-82
Rabbit polyclonal anti-ATOH1	invitrogen	PA5-29392
Rabbit polyclonal anti-ATOH1	proteintech	21215-1-AP

Panels k and l: **The EdU-based lineage tracing data is unclear and confusing.** When after injury was EdU given and when were retinas harvested for analysis. **A timeline of the experiment would help.**

Re: The timeline for sample preparation is provided in panel (a) and the time points of the samples are exhibited in the upper left corner of each figure panel. We added the panels (i.e., b-e for 2 dpi and f-l for 4 dpi), which each EdU labeling was applied.

Finally, based on this figure (panels k and l) it is stated in the text (page 7 top paragraph) that: "Consequently, the regeneration of retinal neurons, identified by their incorporation of EdU and loss of MG-specific gfap-EGFP expression but gain of cell type-specific marker expression, was diminished in the injured zebrafish retinas injected with FLAG-Prox1".. However, the data seems to indicate that **there is no significant effect of Flag-Prox1 on regeneration of most retinal neuron types in the retina as indicated in panels k and l.** **Please clarify.**

Re: We apologize for any confusion caused by the previous presentation. We have changed y-axis values of the graph in Extended Data Fig. 7k to show those in a log scale. We believe this version reveals the differences more clearly. The revised graph clearly shows a decrease in EdU(+) retinal neurons in the MNU+FLAG-Prox1 samples compared to controls, suggesting that FLAG-Prox1 does indeed inhibit retinal regeneration.

We also replaced the graph in Extended Data Fig. 7l with that showing identities of EdU(+) cells relative to MNU + FLAG samples. This version exhibits not only the composition of EdU(+) cells in each sample but also the decrease of EdU(+) cells in FLAG-Prox1-injected samples.

22. Page 8 bottom paragraph: Despite the Hes1 expressing cluster 20 in the scRNAseq data being a relatively small fraction of the whole, HES1 staining in figure 3k show extensive high expression of HES1 throughout the retina. How can this discrepancy be explained?

Re: The cluster 20 represents only about 2.7% of total retinal cells and 14.2% (136/952) of the total Hes1-expressing cell population in the injured *Prox1^{fg/fg};Chx10-CreERT2* mouse retina (Fig. 5d,e). Thus, the majority of Hes1-positive cells in the retina are not RPCs but MG.

It should be noted that Hes1 is expressed at low levels in resting MG and then declines after the retinal injury in wild-type retina (Fig. 5f,g). In contrast, in the *Prox1^{fg/fg};Chx10-CreERT2* mouse retina, Hes1 expression is elevated. The results suggest that the elevated Hes1 might represent the cluster #20 RPC population.

23. Page 9, first full paragraph: Many of the genes being shown in figure ED12 have deceptively low percentage expression. Some percentage/absolute cell expression as in figure 3 could be helpful.

Re: As the reviewer indicates, the genes shown in Extended Data Fig. 12 have low percentage expression due to the high diversity of retinal cell types. To address this, we included the violin plot graphs, which also show the numbers of cells expressing the genes (please see the top rows of Extended Data Fig. 12).

24. Page 9, bottom paragraph: figure 3k,l,m experimental timelines are missing.

Re: We have added the experimental timeline in revised Fig. 6a to address this concern.

25. Page 9, bottom paragraph: The authors state that PROX1 blocks the conversion of MG to proliferative MGPCs; however, the authors have not characterized the resulting proliferative cells enough to state that MGPCs have been established based on PROX1 disruption. Also, the authors state that PROX1 suppresses the expression of target genes such as Notch1 and Ccnd1. While the authors have shown that PROX1 disruption leads to increased Notch1 and Ccnd1 expression. The authors have not established a direct, targeted effect between Prox1 and the downstream genes. Word choice is important here.

Re; We agree with the reviewer's comment. The statement regarding Prox1 blocking the conversion of MG to RPCs was intended as a hypothesis rather than a conclusion. We have revised the text to reflect this, stating: "...exogenous Prox1 might block the conversion of MG to RPCs potentially by suppressing the expression of target genes such as Notch1 and Ccnd1 (highlighted in page 11)." This revision acknowledges the speculative nature of the relationship between Prox1 and these downstream genes.

26. Page 11, top paragraph: Figure 4h is confusing. Even though the y axis label states EdU+Tdtomato+ cell counts, Tdtomato- measures are included in the graph. These graphs should be separated, or the labels should be corrected.

Re: We have corrected the y-axis label to "EdU(+) cells" to accurately reflect the

data presented.

27. Page 11, top paragraph: This is an extension of point #1. **Around 1 EdU+ cell co-labeling with rhodopsin per $2 \times 10^4 \mu\text{m}$ lead to a significant increase in visual acuity.** Is this expected?

Re: Regarding the relatively low number of EdU+ rPRs in AAV2-anti-Prox1-infected rd10 mouse retina, please refer our answers to the reviewer's comment #1.

The significant increase of visual acuity could be explained by the partial recovery of rPR and their outer segments (OS). From the suggestion of the reviewer #3, we have compared transmission electron microscopic (TEM) images of the AAV2-aProx1-infected *Pde6b^{rd10}* mouse retina with those of AAV2-Ctrl Ab-injected samples. The OS structures were identified in AAV2-aProx1-infected mouse retinas, whereas it was absent in AAV2-Ctrl Ab-injected mouse retinas (Reviewer #1's inspection fig. 8; same data as Reviewer's inspection fig. 1). However, the OS numbers are still significantly less than those of wild-type mouse retina. We will provide the results for the reviewer's inspection only, since those are obtained from only one mouse for each sample.

[figure redacted]

Reviewer #1's inspection fig. 8. Partial recovery of photoreceptors in *Pde6b^{rd10/rd10}* mice infected with AAV2-aProx1. Ultrastructure of subretinal spaces of P39 *Pde6b^{+/+}* wild-type mice, *Pde6b^{rd10/rd10}* mice, and *Pde6b^{rd10/rd10}* mice infected with AAV2-aProx1 at P10 were examined by transmission electron microscope. Not only the photoreceptors in the outer nuclear layer (ONL) but the photoreceptor outer segments (OS) were lost significantly in *Pde6b^{rd10/rd10}* mice in comparison to age-matched *Pde6b^{+/+}* wild-type mice. The photoreceptors in the ONL and the OS in the subretinal space between RPE and ONL were observed at significant numbers in *Pde6b^{rd10/rd10}* mouse retina, although those were not recovered to normal numbers seen in *Pde6b^{rd10/rd10}* mouse retina. MV, microvilli of RPE.

28. Page 11, bottom paragraph: **Figure ED15a, c** denotes the timeline of the extended EdU rd10 experiment, but **why are the AAV injection times, tamoxifen injection times, and EdU injection times so much different than anything else**, especially to the comparable rd10 normal experiment of figure 4f? In addition, the analysis timing is given as a range of days. Are the results a mixture of these date ranges or a specific date in these ranges? **If mixtures are used, why are they grouped together rather than being used as a time course?**

Re: In Extended Data Fig. 16b, we have adjusted the timelines for AAV, tamoxifen, and EdU injections to align with those used in Extended Data Fig. 16a. This ensures consistency across experiments. The range of days in the analysis represents a time window within which the experiments were conducted. We grouped these data to account for variations and to provide a more comprehensive view of the results over the range of time points.

29. Page 11, bottom paragraph: in figure 5a and c, the graphs show a steady decline in PROX1 measures while the PROX1 levels across different time points in the pictures appear to be similar. **Maybe other pictures could better represent these differences across time points.**

Re: We have replaced the images to better represent the differences in Prox1 levels

across time points, ensuring that the visual data matches the quantitative graphs more accurately.

30. Page 12 top paragraph: Returning to point #2, the timeline given in figure 5c appears different from everything else. **Why is so much time given between AAV injection and sacrifice without any EdU tracing** even though the desired outcome is measuring the overall proliferation and lineage?

Re: Replication-incompetent AAV needs to infect non-dividing cells to ensure stable gene expression without dilution by repeated cell division. This process typically takes more than two weeks to express the gene fully after infection. Therefore, we performed AAV infection at P60, coinciding with the onset of photoreceptor degeneration in Rp1(m/m) mice. We measured the effects by tracking vision changes and collected retinas at P180 (6M). To evaluate proliferation and lineage, we tracked retinal cells born within a week while minimizing interference from the degeneration of newborn cells.

Why are so few EdU injections given, and all right at the end? Assays of neuronal differentiation 5-9 days post EdU injection may not be sufficient time for proliferating cells to differentiate and it would not reveal if these cells survived in the retina for prolonged periods. **Additional time points post EdU should be explored as should survival of regenerated neurons.**

Re: Respecting the reviewer's suggestion, we extended EdU injections to 4 weeks to capture a broader regenerative window. Consequently, the revised data show an increase in the number of EdU-labeled cells, now presented in updated Fig. 8, d-f.

Also, are regenerated neurons functional?

Re: Assessing the functionality of EdU+;tdTom+ MG-derived photoreceptors in making functional neural circuit is indeed complex and requires single-cell recording, which is beyond our current expertise. Instead, we evaluated retinal function using electroretinogram (ERG) measurements. Results in Extended Data Figs. 15 and 18 show significant increases in ERG a- and b-wave amplitudes, indicative of PR and BC activity, respectively, in Pde6b(rd10/rd10) and Rp1(m/m) mice treated with AAV2-anti-Prox1 compared to controls. Additionally, immunostaining for cone opsin and G0alpha in ON cone BCs in AAV2-anti-Prox1-treated retinas (Extended Data Fig. 19) suggests functional recovery related to the newborn neurons.

31. Page 12 top paragraph: **The methods used to obtain visual acuity measurements as in figure 5f, while given in materials and methods, should be mentioned in the main results body.**

Re: We have included a summary of the methods for visual acuity measurements in the main results section (page 25).

32. Page 15 bottom paragraph: The "area" measurement given in figure ED18c, while relative, should have some explanation behind it. **Is this a measurement of pixel density, intensity, or a combination thereof?**

Re: The "area" measurement in Extended Fig. 19c (ED19c) represents the pixel count of overlapping signals. We have updated the y-axis label in the graph to reflect this and provide a clearer explanation of the measurement.

33. Three different Prox1 antibodies and 2 different Sox2 antibodies are indicated in Table 1. Please **indicate which antibodies were used in each experiment and why they were chosen.**

Re: We have updated Supplementary Table S1 to include details on the specific antibodies used in each experiment, along with the reasons for their selection. This information clarifies the choice of antibodies based on their performance in preliminary tests, specificity, and suitability for different experimental conditions.

34. In the Discussion it is proposed that Prox1 depletion in Muller glial cells stimulates their transition to a progenitor via Notch signaling and these progenitors then proliferate; however **in the zebrafish retina Notch signaling inhibits injury-dependent Muller glia proliferation and in the mouse retina it was reported in a biorxiv preprint from Hoang lab** (Robust reprogramming of glia into neurons by inhibition of Notch signaling and NFI factors in adult mammalian retina) that Notch also inhibits Muller glia cell reprogramming and proliferation in the mouse retina. **This information should be included in the Discussion.**

Re: Notch signaling indeed plays a pivotal role in retinal regeneration. Notch signaling must be initially activated to enable the reprogramming of MG into RPCs. However, once reprogramming is initiated, continued Notch activation can inhibit further progression, such as cell cycle exit and differentiation into retinal neurons. This may be why MG-to-RPC reprogramming is often unsuccessful in the mammalian retina if Notch signaling remains high.

In our study, we utilized the approaches that deplete Prox1 in MG, potentially allowing for reprogramming these cells into RPCs by relieving the inhibitory effects of Prox1 on Notch1 expression. We did not manipulate Notch signaling stably, different from many other studies that overexpress Notch activators or inhibitors in MG. Therefore, the following events that include a feedback inhibition of Notch signaling occur to allow cell cycle exit and neuronal differentiation of the RPCs.

We have revised the Introduction and Discussion to incorporate this information and included the relevant references to provide a comprehensive view of how Notch signaling and Prox1 interact during retinal regeneration.

Reviewer #2 (Remarks to the Author):

In this manuscript Lee, Kim and colleagues investigate the role of Prox1 in limiting the neuro-regenerative potential of Müller glia (MG) following retinal injury. **This study represents a considerable amount of work with hypotheses tested in multiple models and mechanistic experiments giving a strong, robust manuscript. The potential for neuroprotection/recovery in retinitis pigmentosa and the further dissection of mechanisms governing MG reprogramming is of importance to the field.**

The following considerations are needed to improve the manuscript:

Q. I agree with the suitability of the heading “Delaying vision loss in early onset retinitis pigmentosa mouse models by blocking Prox1” but find that some of the description of the results does not match this heading and somewhat oversells the magnitude of effects e.g. **“Significant numbers of EdU-labeled newborn Rhodopsin-positive rPRs were found in the ONL of AAV2- α Prox1-infected *Pde6b^{rd10/rd10};Glast-CreERT;R26^{tdTom/+}* mouse retinas (Fig. 4f,g)”. This does not appear to be supported by any quantification.**

Re: We thank the reviewer for the feedback. As the reviewer indicates, the mean number of EdU(+);tdTom(+);Rhod(+) rPRs in *Pde6b^{rd10/rd10};Glast-CreERT;R26^{tdTom/+}* mouse retinal area is about 4 (combining tdTom(+) and tdTom(-) cells) (Fig. 7h), and it was significantly lower than that observed in MNU-injured zebrafish retina, which are approximately 28 (Extended Data Fig. 7k). Thus, we have revised the description as follows: “EdU-labeled newborn Rhodopsin-positive rPRs were observed in the ONL of AAV2- α Prox1-infected *Pde6b^{rd10/rd10};Glast-CreERT;R26^{tdTom/+}* mouse retinas, whereas no such cells were observed in AAV2-Ctrl Ab-infected mice (Fig. 7g,h)” (highlighted in page 12).

These findings suggest that blocking Prox1 transfer alone may be insufficient to fully restore regenerative capacity to that observed in zebrafish. Additional events, such as Notch inhibition or Yap/Taz activation following post-reprogramming, may be needed to promote robust MG or MGPC proliferation. This consideration has been incorporated into the Discussion (highlighted in page 16), and will be a focus of our future studies.

Similarly, **the ONL thickness increase and scotopic a-wave recovery are extremely modest, rather than “remarkably elevated”.** The language describing the results in this section should be more carefully considered.

Re: We have removed “remarkably” in the expression (last paragraph in page 13, highlighted).

It is unsurprising that the effect is very limited given the harsh degenerative environment of the model and the persistence of the rd10 mutation. This reality is encapsulated nicely in introductory sentence to the following section.

2. The number of mice used for visual acuity in Figure 5G is very low (n = 2).

Re: During the revision, we expanded the mice and added the data to have the number of mice up to 6 in the revised Fig. 8h.

Treated mice only go to 8 months of age. While at this time-point there is complete visual recovery, **it is unknown if this persists or would diminish over time as in the untreated mice which show further decline from 11 months.** It should be noted that with the current data, a conclusion on the long-term benefits of this treatments cannot be made.

Re: We appreciate the reviewer's concern regarding the premature conclusions about the long-term benefits of this treatment. To address this, we extended the monitoring period to 14 months of age in *Rp1^{m/m}* mice injected with AAV-anti-Prox1 at 6 months of age. Our data show that a single injection successfully restores and maintains visual function until 11 months of age (Fig. 8h). However, a decline in visual acuity was observed after 11 months (Fig. 8h). A similar trend was noted in 9-month-old *Rp1^{m/m}* mice infected with AAV2-anti-Prox1 at 2 months of age (Fig. 8g).

We hypothesize that this limitation may be related to the silencing of the EF1A promoter used to drive antibody gene expression in our AAV vector, but not due to the depletion of MG that are expected to regenerate retinal cells (Extended Data Fig. 20). As we wrote in the manuscript (page 14), the EF1A promoter is prone to silencing during transgene expression *in vivo* (Battulin et al., 2022, Transgenic Res 31, 525-535; Seita et al., 2019, Biol Reprod 100, 1440-1452; Eun et al., 2020, PLoS One 15, e0233784), although it is used widely to express transgene in the retina transiently (Jüttner et al., 2019, Nature Neurosci 22, 1345-1356). To overcome this limitation, we are currently redesigning the promoter to achieve sustained anti-Prox1 expression in the retina, which we anticipate will enhance the long-term efficacy of the treatment.

3. Extended data 7K. Number of EDU+ Sox2+ cells is significantly reduced with FLAG-Prox1 but this is a limited reduction of roughly 30%. Data in 7I suggest that the proportion of EDU+ Sox2+ cells is increased, driven by an equally large loss of EDU in other cells (red). What is the identity of these cells, suggest that Prox1 limits proliferation of more than just MG in this context?

Re: The graph shows composition of EdU+ cells in each experimental group. Therefore, it cannot reflect actual decrease of EdU+ cell number in FLAG-Prox1 injected fish retinas in comparison to FLAG-injected groups. In the revised Extended Data Fig. 7I, we replaced the graph, which exhibit both the numbers and composition of EdU+ retinal cells.

4. The author's hypothesis that chondroitin sulfate proteoglycans facilitate Prox1 transfer could be tested by intravitreal injection of Chondroitinase ABC after NMDA injury in mice and quantify the effects on Prox1 transfer to MG.

Re: Following to the reviewer's suggestion, we injected Chnase ABC to the vitreous of MNU-injured mouse eyes and investigated Prox1 distribution in the mouse retina. Our data show Prox1 level in MG of MNU-injured mouse retina was decreased by the injection of Chnase ABC (Reviewer #2's Inspection fig. 1). We, however, do not include the results in the paper, but provide those only for the reviewer's inspection. We will report these in a separate paper together with the results obtaining in the mice defective to chondroitin sulfate synthesis in MG.

[figure redacted]

Reviewer #2's inspection fig. 1. Prox1 transfer to MG is sensitive to chondroitin sulfate glycosaminoglycan. (a) P30 *Glast-CreER;R26^{tdTom/+}* mice were injected with Tam to activate CreER and express tdTom reporter in MG cell lineage. In these mice, MNU retinal injury was applied by intraperitoneal (IP) injection, followed by intravitreal (IVT) injection of PBS or chondroitinase ABC (ChABC; 0.25units/eye). **(b)** The retinal sections were prepared from the mice for the immunostaining of Prox1 in Sox2-positive MG. Proper actions of CreER and ChABC were determined by visualizing tdTom reporter and AlexaFluor647-labeled wheat agglutinin (WFA), which bind specifically to chondroitin sulfates. **(c)** Relative Prox1 intensity in MG to that in Vehicle(IP)+PBS(IVT) injection group is shown in the graph.

5. Introduction is missing a clear hypothesis or aim, not clear what knowledge gap is being addressed from this work for the general reader

Re: We apologize for short Introduction. We added background and hypothesis of the study in Introduction of revised manuscript.

Minor considerations

1. What is the significance of the increase of EdU-labeled microglia in the injured retinas of Prox1^{fg/fg};Chx10-CreERT2 mice? Does MG proliferation result in reduced MG phagocytic capacity and a resultant increase in microglia to fill this role?

Re: The numbers of EdU-labeled microglia were elevated in FLAG-Prox1-Prox1^{fg/fg};Chx10-CreERT2 mouse retina, whereas those are unchanged in Prox1^{fg/fg};Chx10-CreERT2 mouse retina, in comparison to Prox1^{fg/fg} littermated mouse retina (Fig. 3f). The elevation suggested that microglial expansion is induced by FLAG-Prox1, which penetrates and accumulates in MG.

2. Pg 20. "resuspended in resuspended in the resuspension buffer"

Re: Thank you for indication. We corrected the sentence in the revised manuscript.

Reviewer #3 (Remarks to the Author):

In this study, the authors demonstrate the novel importance of Prox1 in rendering retinal MG incapable of reentering the cell cycle and becoming MGPCs. Prox1 accumulates in MG in retinal degenerations and was seen to originate from neighboring neurons, in this case they primarily studied Prox1 from bipolar cells. Deletion of bipolar cell Prox1 gene or sequestration of extracellular Prox1 protein by an antibody induced the reprogramming of MG into proliferative retinal progenitor cells in injured retina. Using these findings, the authors successfully delayed vision loss in mouse models of RP. **It is a very interesting study that should be of significance, and well written. The finding of intercellular Prox1 transfer is novel and the delay in vision loss is striking, particularly in the late onset model. I only have a few points.**

(1) The ending of the **Introduction feels very abrupt and seems like it needs a hypothesis or aim statement.**

Re: We apologize for short Introduction. We added background and hypothesis of the study in Introduction of revised manuscript.

(2) Supp Fig 1d has incorrect figure legend colors.

Re: We apologize for the insufficient explanation. Corresponding images for MNU injury in the graph are provided in Fig. 1a. We added the explanation in the figure legend.

(3) It would be nice to have seen **electron microscopy images to see how much of the photoreceptor structure is preserved in the AAV2-aProx1 infected RD10 mice.**

Re: Respecting the reviewer's suggestion, we have compared transmission electron microscopic (TEM) images of the AAV2-aProx1-infected *Pde6b^{rd10/rd10}* mouse retina with those of AAV2-Ctrl Ab-injected samples. The rPR outer segment (OS) structures were identified in AAV2-aProx1-infected mouse retinas, whereas it was absent in AAV2-Ctrl Ab-infected mouse retinas (Reviewer #3's inspection fig. 1). However, the OS numbers are still significantly less than those of wild-type mouse retina. We will provide the results for the reviewer's inspection only, since those are obtained from only one mouse for each sample.

[figure redacted]

Reviewer #3's inspection fig. 1. Partial recovery of photoreceptors in *Pde6b^{rd10/rd10}* mice infected with AAV2-aProx1. Ultrastructure of subretinal space of P39 *Pde6b^{+/+}* wild-type mice, *Pde6b^{rd10/rd10}* mice, and *Pde6b^{rd10/rd10}* mice infected with AAV2-aProx1 at P10 was examined by transmission electron microscope. Not only the photoreceptors in the outer nuclear layer (ONL) but the photoreceptor outer segments (OS) were lost significantly in *Pde6b^{rd10/rd10}* mice in comparison to age-matched *Pde6b^{+/+}* wild-type. The photoreceptors in the ONL and the OS in the subretinal space between RPE and ONL were observed at significant numbers in *Pde6b^{rd10/rd10}* mouse retina, although those were not recovered to normal numbers seen in *Pde6b^{rd10/rd10}* mouse retina. MV, microvilli of RPE.

(4) In Fig 2e the authors have counted the number of Sox2+/Edu+ cells but what proportion of MG become Edu+. I think this proportion is important to show, to highlight the effectiveness of eliminating Prox1. **What proportion of MG become reprogrammed and turn into other cell types?**

Re: Given the expression of Sox2 both in MG and RPC, we cannot determine the

portion of RPC among total Sox2-positive cells. Instead, we could determine RPC population in scRNA-seq analyses of *Prox1^{fg/fg};Chx10-CreERT2* mouse retinas. Our scRNA-seq data show that 7.6% (2.7% / 36.2%) of MG marker expressing cells, which include the cluster #1, 3, 7, and 20, exhibited RPC markers in *Prox1^{fg/fg};Chx10-CreERT2* mouse retinas, whereas those are just 0.27% of MG population in *Chx10-CreERT2* mouse retinas (Extended Data Fig. 11). Therefore, our data suggest that approximately 8% of MG were converted to RPC in *Prox1^{fg/fg};Chx10-CreERT2* mouse retinas.

On a similar note, **what proportion of Edu+ cells become the other different cell types?** (e.g., what proportion of Edu+ cells colocalize with Iba+ or PKCa).

Re: The EdU-labeled cells in MNU-injured *Prox1^{fg/fg};Chx10-CreERT2* mouse retinas are composed of Sox2+ MG or MGPCs (69.8%) and Iba1+ microglia (30.2%) (Fig. 3g). We could not find other cell population among EdU-labeled cells in the mouse retinas.

Reviewer #1 (Remarks to the Author):

Lee et al., did a good job in revising their Prox1 paper. However, a relatively small number of comments remain to be addressed.

Major adjustments to the paper

1. Transdifferentiation vs neuropreservation: Although the authors demonstrate little or no changes in neuroprotection upon Prox1 suppression, they do not demonstrate a lack of transdifferentiation as claimed. Selecting tdT+ cells that are Sox2- and EdU- for this analysis may result in a bias in your analysis. It is clear from the research papers reported from the Reh and Blackshaw labs that EdU+ and Sox2+ MG can transdifferentiate. Just because a MG cell undergoes cell division does not mean it is a neuronal progenitor. Furthermore, Notch inhibition and NFi suppression are only a couple of ways to stimulate transdifferentiation. Other ways include Ascl1 overexpression and Trichostatin A treatment as shown by the Reh lab. Thus, there are likely many ways to stimulate transdifferentiation. I think the authors should rely more on their scRNAseq data sets to confirm or refute MG transition to a proliferating retinal progenitor state and to inform the reader as to the extent this transition occurs by comparing to scRNAseq data previously obtained using RNA from the developing retina (Hoang, Blackshaw data).

Re: We appreciate the reviewer's insightful comment regarding the potential underestimation of transdifferentiation and the challenges associated with distinguishing between regeneration and transdifferentiation. In this paper, we identified EdU incorporation in terminally differentiated retinal neurons as a marker for regeneration of the neuron from MG via MGPCs. However, as the reviewer points out, EdU-labeled neurons could also be transdifferentiated from newborn MG that have not been fully reprogrammed into MGPCs. In light of this, we have refrained from drawing definitive conclusions about regeneration versus transdifferentiation. As requested, we have revised the Discussion section (page 15, green highlight) to include our updated interpretation of the origin of the EdU-labeled retinal neurons, as well as the potential limitations of our current approach.

2. Regarding Chx10 promoter: although you report in your rebuttal that Chx10 promoter is expressed in ~20% of MG, you also state in the text (line 182) it is BC specific. "...CreERT2 selectively in the BC population in a Tam-dependent manner ..." The text should be edited for accuracy.

Re: We appreciate the reviewer pointing out this inconsistency regarding the specificity of the Chx10 promoter. To ensure accuracy, we have revised the text (page 7, highlight), replacing "selectively" with "predominantly." This change better reflects the observed expression pattern and avoids overstating its specificity to BCs.

In addition, because Prox1 RNA is expressed in MG, the expression of CreER in MG, even at low levels, is likely to reduce this expression thus confounding the conclusion that MG Prox1 is derived from BP cells.

Re: We demonstrated that there is no detectable decrease in Prox1 level in MG of Prox1^{fg/fg};Glast-CreER mouse retinas (Fig. 2e,g). This observation suggests that the decrease of Prox1 in MG of Prox1^{fg/fg};Chx10-CreERT2 mouse retinas is unlikely resulted from the loss of endogenous Prox1. Thus, we concluded that this reduction might be due to the loss of Prox1 in BCs, which secrete Prox1 that is subsequently transferred to MG.

3. Reviewer #1's figure 4 as well as the explanation of the extracellular nature of Prox1a/Prox1 should be included in the main body. Same for Reviewer #1 figure 5.

Re: We have included the two figures as Extended Data Fig. 8 and Extended Data Fig. 17.

Regarding Extended Data Fig. 8, we have revised the manuscript to include the following explanation (pages 8–9, green highlight): "Alternatively, we also aimed to disrupt Prox1 transfer to MG by sequestering it in the extracellular space using anti-Prox1 antibody (Fig. 4a). The rabbit polyclonal anti-Prox1 antibody that was injected into the intravitreal space of MNU-injured Glast-CreERT;R26^{tdTom/+} mouse eyes could reduce Prox1 level in MG, suggesting the blockage of intercellular transfer of Prox1 by the antibody (Extended Data Fig. 8)."

Regarding Extended Data Fig. 17, we included the following explanation (page 13, green highlight): "In contrast, no visual recovery was noted in Pde6b^{rd10/rd10} mice infected with AAV2- α Prox1 at P60 (Extended Data Fig. 17), indicating that AAV2-anti-Prox1-induced retinal restoration requires early intervention, prior to the completion of retinal degeneration".

4. Regarding Reviewer#1 comment 18 and response highlighted in text on page 8. ED6 indicates

the FLAG-peptide remains in the vitreous of the injured retina while the FLAG-Prox1 can move from the vitreous into the injured retina. What is the explanation for this different behavior of the 2 proteins? If the FLAG-peptide never enters the retina, it may not be a good control for the FLAG-Prox1 that does enter the retina.

Re: While FLAG peptides can diffuse into the retina from the vitreous, they lack specific affinity for retinal cells. In contrast, FLAG-Prox1 exhibits an intrinsic ability to interact with and become entrapped by retinal cells, including MG. Due to the lack of cellular affinity, FLAG peptides do not remain stably within the retina and are therefore not detectable at significant levels. We have clarified this distinction in the revised text on page 8 (green highlight).

Minor adjustments

1. Figure 1d is not cited in the paper

Re: Thank you for your comment. The relevant data is now cited in the revised manuscript, which can be found on page 5, marked with a green highlight for clarity.

2. Ext Fig 9c is not cited

Re: We thank the reviewer for pointing this out. The relevant data is cited as Extended Data Fig. 10c in the revised manuscript (page 9, green highlight).

3. No quantification of EdU+ or Ascl1+ cells at 4dpi in the zebrafish after Prox1 protein injection and MNU injury.

Re: Ascl1-expressing cells were not detectable at significant numbers in zebrafish retinas at 4dpi (please find the results in the following for the reviewer's inspection only). This low frequency of Ascl1-positive cells hindered reliable quantification, particularly when assessing the inhibitory effects of FLAG-Prox1 on MGPC formation from MG.

[figure redacted]

Ascl1 expression in MNU-injured zebrafish retina (figure for reviewer's inspection only).
Retinal injury was induced in 6-month-old *gfap:zebrafish* through intraperitoneal injection of MNU, followed by immunostaining to assess

	Ascl1 distribution in the injured fish retinas. The bottom three row displays single focal images, magnified from the boxed areas in the stacked images shown in the top row. MGPC identities of Ascl1-positive cells were determined by EGFP co-expression. Yellow arrowheads point Ascl1;EGFP double-positive cells.
--	---

4. Fig 6a: Are uninjured bipolar cells less in number compared to the injured, or is this a case of relative abundance?

Re: Fig. 6a depicts the timeline for SMART-seq sample preparation and does not include images or quantitative data on bipolar cell abundance. Therefore, it does not address the relative or absolute numbers of injured versus uninjured bipolar cells.

If the question pertains to Fig. 5a, the observed changes likely reflect a relative increase in bipolar cells due to photoreceptor degeneration in injured mouse retinas.

5. Fig7h: The existence of EdU+Tdt- cells should be explained with the Glast-Cre quantification provided in the review where only ~60% of Sox2+ MG were labeled with Tdtomato.

Re: Thank you for raising this point. We have addressed this in the revised manuscript by adding the sentence: “Notably, due to the incomplete penetrance of Glast-CreERT activity in MG, the tdTom-negative, EdU-labeled retinal neurons observed in these retinas could also have been derived from MG” (page 12, green highlight).

6. Ext fig 8a: Why do separate staining of the Prox1a-Prox1 pulldown using Prox1 vs HA antibody produce drastically different sized bands?

Re: In our Western blot (WB) analysis, the anti-Prox1 antibody detects Prox1 protein, which has a molecular weight of 83 kDa but migrates to approximately the 95 kDa position in SDS-PAGE. In contrast, the anti-HA antibody detects the α Prox1-6His-HA scFv protein, which has a molecular weight of approximately 35 kDa. As a result, these two proteins are clearly distinguishable and are observed at different positions in the WB images.

Q. Ext fig 8: In the legends “they receive the HP from HP” requires clarification of both the sentence and HP.

Re: The original legend reads: "..., suggesting that they received the HP from HP;EGFP-positive neighboring cells." To enhance clarity, we revised it as: "..., suggesting that they received the HP from HP;EGFP double-positive neighboring cells."

8. Page 3 line 75 to 77: "Notch signaling is also activated... neuronal differentiation." The line appears confusing in stating that Notch is activated but should go down. Clarification is required in stating differential Notch factor expression is required for zebrafish MG regeneration.

Re: The sentence in the revised manuscript already conveys this meaning: "...however it should be suppressed for MGPC proliferation and their subsequent neuronal differentiation."

9. Page 4 line 83 to 85: "further support this process... TAZ" Suggest replacing "which suppresses yes-associated protein" with "leading to the disinhibition of yes-associated protein" to avoid confusion in this phrase either modifying "the inactivation" or "Hippo signaling."

Re: We thank the reviewer for the suggestion. We have revised the sentence to: "In addition to these external cues, changes in intracellular pathways that suppresses yes-associated protein (Yap) and WW domain-containing transcription regulator 1 (TAZ) can also support this process" (page 4, green highlight).

10. Page 16 line 429 Extra: "in" or replace with "in the mouse retina" in the middle of the sentence "the number of EdU-labeled cells in remains lower..."

Re: We thank the reviewer for pointing out this error. We have removed the extra "in" in the revised manuscript (page 16, green highlight).

11. Page 21 line 580 to 587: The production and injection of control FLAG peptide was not described in the methods.

Re: We added the methods preparing FLAG peptides in Methods (page 22 – 23, green highlight)

12. Page 22 line 589 to 597: AAV plasmids, production, and controls used were not described in the methods.

Re: We added the methods for AAV production in Methods (page 23, green highlight)

13. Page 5 line 130: missing "is" in sentence "where selective PR degeneration induced by DTA."

Re: We thank the reviewer for the correction. We added "is" in the revised manuscript (page 5, green highlight).

14. Page 9 line 247: "We also observed a distinctive lineage from to the BC..." Requires some editing.

Re: We thank the reviewer for the correction. We removed "from" in the revised manuscript (page 10, green highlight).